# Identification of enzymes that have helminth-specific active sites and are required for Rhodoquinone-dependent metabolism as targets for new anthelmintics

**Margot J. Lautens**[ID][�], **June H. Tan**[ID][�], **Xènia Serrat**[ID], **Samantha Del Borrello**[ID], **Michael R. Schertzberg, Andrew G. Fraser**[ID]*

The Donnelly Centre, University of Toronto, Toronto, Ontario, Canada

start These authors contributed equally to this work.
* andyfraser.utoronto@gmail.com

## Abstract

Soil transmitted helminths (STHs) are major human pathogens that infect over a billion people. Resistance to current anthelmintics is rising and new drugs are needed. Here we combine multiple approaches to find druggable targets in the anaerobic metabolic pathways STHs need to survive in their mammalian host. These require rhodoquinone (RQ), an electron carrier used by STHs and not their hosts. We identified 25 genes predicted to act in RQ-dependent metabolism including sensing hypoxia and RQ synthesis and found 9 are required. Since all 9 have mammalian orthologues, we used comparative genomics and structural modeling to identify those with active sites that differ between host and parasite. Together, we found 4 genes that are required for RQ-dependent metabolism and have different active sites. Finding these high confidence targets can open up *in silico* screens to identify species selective inhibitors of these enzymes as new anthelmintics.

## Author summary

Soil-transmitted parasitic worms infect over a billion humans worldwide. In recent years, there has been increasing resistance to existing drugs that are used to treat parasitic infections and this has accelerated the need to develop new classes of drugs. Parasitic worms survive in the low oxygen environment of the host gut, using a special electron carrier, rhodoquinone. Since rhodoquinone is not used or made in humans, enzymes that use or make rhodoquinone are good targets for the development of new drugs. Here, we first identify genes that are important for parasites to carry out these rhodoquinone-dependent processes. Since many of these genes are also present in the human genome and we only want to target the parasite version, we then looked for sequence differences in these genes between parasites and humans to identify enzymes with parasite-specific active sites. These sequence differences may allow for drugs which bind only to the parasite-version of

**Data Availability Statement:** All relevant data are within the manuscript and its Supporting information files.

**Funding:** AF received a grant from the Canadian Institutes of Health Research (CIHR) <https://cihr-irsc.gc.ca/e/193.html>, grant #501584 and 5003009. XS is supported by the European Molecular Biology Organization (EMBO ALTF 387-2021). The funder had no role in study design, data collection and analysis, decision to publish, or preparation of the manuscript.

**Competing interests:** The authors have declared that no competing interests exist.

the enzymes. From this, we have identified 4 high-confidence targets that can be used to guide the design of new anti-parasitic drugs.

## Introduction

Soil-Transmitted Helminths (STHs) such as *Ascaris*, whipworm and hookworm are parasites that infect over a billion humans and a large proportion of livestock [1–4]. Anthelmintic drugs such as ivermectin [5,6] levamisole [7] and benzimidazoles [8] are listed amongst the WHO Essential Drugs [9] and have provided frontline treatment for these pathogens for decades. However, anthelmintic resistance is widespread in livestock [10–13] and, in humans, efficacy for specific drugs have decreased in certain regions [14–16] and few classes of anthelmintics currently exist. There is thus an urgent need for new classes of anthelmintics to combat these major pathogens.

STHs have complex and diverse life cycles but share a key feature that can be exploited for the development of anthelmintic treatments. STHs go through profound changes to adapt to the host environment and some of these involve STH-specific biology and machineries. These STH-specific molecular events are excellent potential targets for new anthelmintics [17]. One of the best studied is the metabolic switch from standard aerobic energy metabolism that uses ubiquinone (UQ) as a critical electron carrier in the mitochondrial electron transport chain (ETC) to an unusual anaerobic metabolism that requires the unusual electron carrier rhodoquinone (RQ) [18–21] in a rewired ETC [22–24]. The key difference is that when UQ is used as an electron carrier, the sole terminal electron acceptor in the ETC is oxygen. When RQ is used, however, it allows electrons to be routed to a number of alternative electron acceptors [23,25,26]. The RQ-dependent rewired ETC allows STHs to continue using the mitochondrial ETC to power ATP synthesis in the absence of oxygen and lets STHs survive in many highly anaerobic niches in the host such as the intestine. This is reviewed in depth in [27].

RQ-dependent metabolism is essential for parasite survival in the host, and it is also a critical difference between parasites and their mammalian hosts. Mammalian hosts do not have RQ—they only ever make and use UQ. STHs must be able to make and use both UQ and RQ and drugs that block RQ synthesis or its use should thus kill the STHs without affecting their hosts. The need to make and use both RQ and UQ has driven a number of STH-specific molecular changes. The best characterised are the changes surrounding the quinone-binding pocket of succinate dehydrogenase, also known as Complex II [28–30]. In aerobic conditions, STH Complex II acts as a dehydrogenase, converting succinate to fumarate and transferring two electrons onto UQ as the entry point into the ETC—this is the same basic function as host Complex II. In anaerobic conditions, however, STHs reverse this reaction: Complex II now acts as a fumarate reductase and the two electrons required to convert fumarate to succinate must come from RQ [18,23]. STH Complex II now acts as a point for electrons to exit the ETC and fumarate rather than oxygen is the terminal electron acceptor. Since STH Complex II uses both UQ and RQ, the quinone-binding pocket of STH Complex II must be able to dock both UQ and RQ whereas the host Complex II only ever binds UQ. UQ and RQ are highly related but differ at position 2 of their benzoquinone rings: UQ has a methoxy group whereas RQ has an amine group [19]. STH Complex II must be able to bind both ring structures and this difference in quinone-binding requirement has led to a number of STH-specific amino acid changes surrounding the quinone-binding pocket of Complex II [31–33]. The difference between the quinone-binding pocket of STHs and the hosts has allowed the identification of several STH-specific Complex II inhibitors and we previously showed that these STH-specific

inhibitors can efficiently kill *C. elegans*, a free-living Clade V nematode that is related to STHs (Fig 1A), when it requires RQ-dependent metabolism to survive [34]. We note that throughout this paper we use *C. elegans* as a model for STHs specifically as regards RQ-dependent metabolism though recognise that there are many biological differences between a free-living nematode and a parasitic helminth. Nonetheless it is the closest related model organism to STHs and the fundamental process of RQ synthesis and RQ dependent metabolism is likely to be largely conserved between these related species.

Complex II is not the sole component of quinone-coupled metabolism known to differ between mammalian hosts and STHs as a consequence of the difference in quinone utilization. We previously showed that STHs are able to make both RQ and UQ whereas hosts make UQ alone because of differences in the core of the key polyprenyltransferase enzyme COQ-2 [35]. COQ-2 is highly conserved and almost all eukaryotes share the same COQ-2 core. However, we found that *C. elegans* use an alternative splicing event to incorporate two key amino acid changes into the core and that STHs show the same AS events [35]. Alternative splicing of two mutually exclusive exons allows *C. elegans* (and likely STH) COQ-2 to be able to use either 4-hydroxy benzoate (4HB) as a substrate for UQ synthesis or 3-hydroxy anthranilate for RQ synthesis—other mammalian hosts can only use 4HB. Thus, the requirement to make or use RQ as well as UQ has resulted in changes in active sites of several key enzymes resulting in differences between parasites and mammalian hosts. These STH-specific active sites make them ideal targets for STH-specific inhibitors which could result in new classes of anthelmintic drugs. We reasoned that other enzymes that are required for RQ-dependent metabolism may have had similar active site changes and thus could also be targets for STH-specific inhibitors. Our goal in this study is to identify such targets and we take a two-step approach to identify parasite proteins as high confidence targets for inhibitors of RQ-dependent metabolism for *in vitro* or *in silico* screens.

We initially assembled a set of rationally-chosen candidate genes that are predicted to act in one of four aspects of RQ-dependent metabolism: how *C. elegans* (and likely STHs) sense the need to switch from UQ to RQ-dependent energy metabolism; how it makes RQ; the quinone-coupled dehydrogenases (QDHs) that directly interact with RQ; and other metabolic enzymes downstream of QDHs that are required for efficient RQ-dependent metabolism (Fig 1B). In our two-step approach, we first experimentally test which of these candidate genes is required for RQ-dependent metabolism in *C. elegans*. We do this using an assay that we previously described that allows us to assess the ability of *C. elegans* to use RQ-dependent metabolism [34]. Given the evolutionary relatedness between *C. elegans* and STHs (Fig 1A), we think it is likely that their closely related orthologues carry out similar roles in RQ-dependent metabolism in parasites and thus are excellent potential drug targets: if we could specifically block their activity without affecting their mammalian host, they should kill the parasite *in vivo*. In our second step, we use computational analyses to determine which of the genes we identify in our experimental testing have helminth-specific changes in druggable regions of the protein— for example, changes surrounding the active site of an enzyme. This second step is critical since RQ-dependent metabolism does not use a dedicated suite of helminth-specific genes but is largely a rewiring of highly conserved genes—indeed almost all of the candidate genes have host orthologues. While most helminth orthologues of human genes show sequence divergence, enzymes often have a high degree of similarity and in particular the active sites tend to be highly conserved. This makes the development of helminth-specific inhibitors very challenging. We therefore combine multi-species sequence alignments with structural modeling to identify which of the potential targets that we identify from our experimental screening have helminth-specific sequence changes at or near the active sites. These would then be the highest priority targets for *in vitro* screens for inhibitors that can block the unusual RQ-dependent

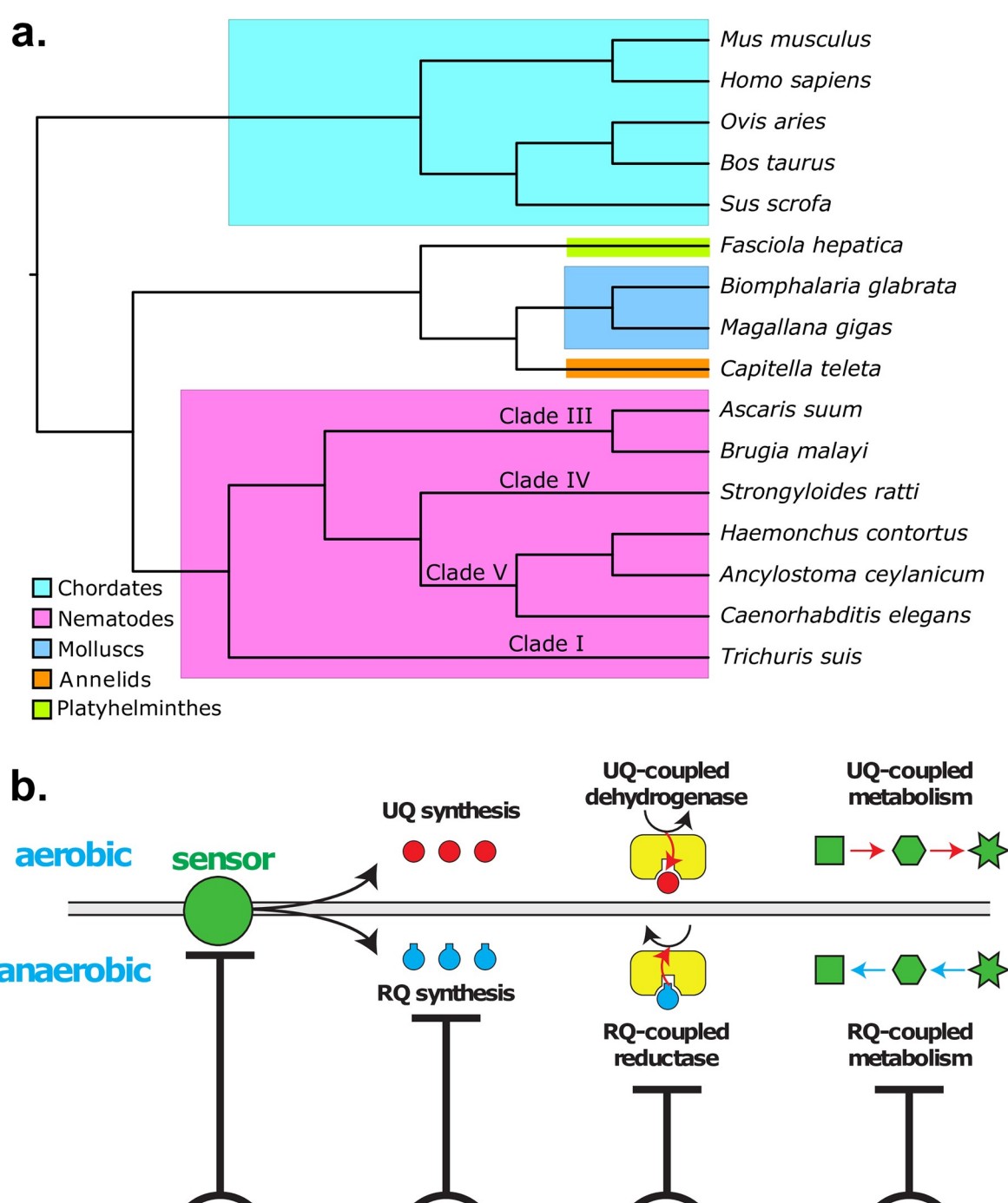

**Fig 1. Identification of potential anthelmintic targets that act in various aspects of RQ-dependent metabolism.** a) Evolutionary tree of the species whose sequences are used in this paper. Species are grouped according to the phylum: chordates (the mammalian hosts, cyan), platyhelminthes (chartreuse), molluscs (blue), annelids (orange) and nematodes (magenta). Helminths are further identified by their clade. The tree was generated with {rotl} [36]. b) Genes involved in regulating four main areas of RQ-dependent metabolism are candidate genes for helminth-specific inhibitors. (1) Genes that act to sense and regulate the switch from UQ to RQ-dependent metabolism as STHs migrate to anaerobic environments in the hosts are potential anthelmintic targets ('sensors') (2) Genes involved in RQ synthesis (3) The Quinone-dependent Dehydrogenases (QDHs) that interact with RQ (4) Other metabolic enzymes downstream of QDHs that are important for the rewiring of metabolic pathways in anaerobic environments are also potential targets.

metabolism that parasitic helminths rely on to survive in their host without affecting any host enzymes.

In this paper, we tested 25 candidate genes for their requirement for RQ-dependent metabolism in *C. elegans*. We found 9 genes to be required for *C. elegans* to carry out efficient RQ-dependent metabolism and suggest that their orthologues in STHs play similar essential roles in RQ-dependent metabolism given their relatedness. These include sensors that regulate the switch from UQ-dependent metabolism to RQ-dependent metabolism, enzymes required for RQ synthesis, and a quinone-coupled dehydrogenase. Using sequence alignment and structural modeling we show that 3 of these have active sites that differ between STHs and their mammalian hosts—these are high confidence targets since the STH-specific active site should allow the development of STH-specific inhibitors that could act as new classes of anthelmintics. We also identify an additional STH-specific target that has no mammalian homologue and that is essential for *C. elegans* survival. We believe that this work provides an excellent platform for *in vitro* and *in silico* screens for compounds that block these key enzymes in STHs without affecting their mammalian hosts.

## Results

### A set of candidate genes with potential roles in RQ-dependent metabolism

We previously developed an image-based assay that allows us to test whether any particular gene is required for RQ-dependent metabolism in *C. elegans* [34,37]. If so, it is a candidate target for anthelmintics since inhibiting its function would kill the STH. In brief outline, we expose *C. elegans* L1 larvae to potassium cyanide (KCN) for 15 hours—KCN prevents oxygen binding to Complex IV of the mitochondrial electron transport chain (ETC) and thus blocks oxidative phosphorylation. As we previously showed [34], this forces *C. elegans* to rely on anaerobic RQ-dependent metabolism and wild-type worms readily survive 15 hours of KCN treatment and rapidly recover full movement after removal of KCN. Worms that cannot make RQ, such as animals with mutations in the gene *kynu-1*, cannot survive and thus show no recovery of movement once KCN is removed [34]. We note that this assay works not just on L1 worms but can also be used to drive *C. elegans* into using RQ-dependent metabolism at other developmental stages (S1 Fig), suggesting our findings should likely be applicable to other life stages. This assay is powerful but does not have sufficient throughput to carry out genome-scale screens and thus we were forced to take a candidate gene approach. Fortunately, much is known about the pathways that may regulate or be required for RQ-dependent metabolism and thus we manually assembled a set of candidate genes to test for their involvement in RQ-dependent metabolism. These genes are all potential targets for anthelmintics and are predicted from previous literature to act in one of 4 different aspects of RQ-dependent metabolism (shown schematically in Fig 1B; genes are listed in Table 1).

First, helminths must sense a requirement to switch from UQ to RQ-dependent metabolism [39] such as a change in environment from an aerobic niche to an anaerobic one. Interfering with this sensor could thus block the ability of parasites to activate RQ-dependent metabolism. Second, in response to the change in environment, parasites alter their quinone synthesis pathways to make more RQ and less UQ. For example, ~14% of the quinone in free-living miracidia-stage *Fasciola hepatica* is RQ, but ~93% is RQ in the adult stage in the host [40]. Blocking RQ synthesis will block RQ-dependent metabolism, and this is a key potential target for anthelmintics. Third, once RQ has been synthesised, it affects metabolism principally by directly docking to the quinone-coupled dehydrogenases (QDHs) that are linked to the mitochondrial ETC [41,42]. If a compound blocks the docking of RQ to the critical QDHs, this should block RQ-dependent metabolism. There are multiple QDHs that link to the ETC

**Table 1. Candidate genes with potential roles in RQ-dependent metabolism.**

| Category | Gene Name | Perturbation | Genotype | Required for KCN-dependent Metabolism | Active Site Changes |
|---|---|---|---|---|---|
| Sensors | aak-2 | Mutation | aak-2(rr48) aak-2(ok524) | + | N/A |
| | hif-1 | Mutation | hif-1(ia4) | 0 | N/A |
| | ire-1 | Mutation | ire-1(zc14);zcIs4 ire-1(ok799) | 0 | N/A |
| | rict-1 | Mutation | rict-1(mg360) rict-1(ft7) | ++ | 0 |
| | sgk-1 | Mutation | sgk-1(ok538) | ± | + |
| | xbp-1 | Mutation | xbp-1(zc12); zcIs4 | 0 | N/A |
| RQ Synthesis | coq-2[a] | Mutation | coq-2(syb1721) | ±± | ++ |
| | kynu-1 | Mutation | kynu-1(e1003) | ±± | 0 |
| | tdo-2 | Mutation | tdo-2(zg216) tdo-2(zg217) tdo-2(zg218) | ±± | ++ |
| | coq-3 | RNAi | N/A | + | 0 |
| | coq-5 | RNAi | N/A | 0 | N/A |
| | afmd-1 | N/A | N/A | N/A | N/A |
| | kmo-1 | N/A | N/A | N/A | 0 |
| RQ Binding | mev-1[b] | Mutation | mev-1(tr357) | ++ | ++ |
| | dhod-1 | RNAi | N/A | 0 | N/A |
| | gpdh-1 | RNAi | N/A | 0 | N/A |
| | let-721 | RNAi | N/A | ++ | ++ |
| | prdh-1 | RNAi | N/A | 0 | N/A |
| | F27D4.1 | RNAi | N/A | + | N/A |
| Metabolism | F36A2.3 | RNAi | N/A | 0 | N/A |
| | men-1 | RNAi | N/A | 0 | N/A |
| | VF13D12L.3.1 | Mutation | VF13D12L.3.1(syb243) | 0 | N/A |
| | pyc-1 | RNAi | N/A | 0 | N/A |
| | mdh-1 | RNAi | N/A | 0 | N/A |
| | icl-1 | Mutation | icl-1(ok531) | 0 | N/A |

Candidate genes are grouped by category and the perturbation(s) is identified for each. Perturbations which resulted in a 50% reduction in movement at the median of all reps are marked with a + and coloured yellow, those which resulted in a reduction in movement of 25% at the median are marked with a ++ and coloured blue. Text is in underlined if the difference was $1.00e{-}02 < p <= 5.00e{-}02$, in italics if it was $1.00e{-}03 < p <= 1.00e{-}02$, in bold if it was $1.00e{-}04 < p <= 1.00e{-}03$ and in bold-underlines if it was $p <= 1.00e{-}05$, all according to Welsh's $t$-test. Active site changes were marked with a + and coloured yellow if S3Det analysis [38] identified residues which were in the vicinity of the active site and with a ++ and coloured blue if those residues were close to the active site and were a significant amino acid change.

[a] Functional assay and structure analysis described in [35].

[b] Functional assay and structure analysis described in [34].

and we must test which are required for RQ-dependent metabolism and thus which are the potential targets. Finally, multiple other metabolic enzymes are required during RQ-dependent metabolism downstream of the interaction between RQ and QDHs [24]. By screening 25 candidate genes in our RQ-dependent metabolism assay, we aim to find a set of genes that are essential for *C. elegans* to carry out anaerobic RQ-dependent metabolism and suggest that their orthologues will act similarly in parasites. We describe the data from these assays on each of the four classes of genes below.

## The switch to RQ-dependent metabolism is not a classic HIF-1-dependent hypoxia response

Parasitic helminths switch from aerobic UQ-dependent metabolism to anaerobic RQ-dependent metabolism when they are exposed to prolonged hypoxia. They must sense this change in environment and blocking the hypoxia response might prevent helminths from switching into anaerobic metabolism. The hypoxia-sensing machinery is thus an excellent potential target for anthelmintics.

The machinery for sensing hypoxia has been well-described in *C. elegans* [43–48] and requires the core hypoxia-regulated transcription factor HIF-1 [43,48] along with its negative regulators EGL-9 [44,49] and VHL-1 [47,50], and the atypical splicing regulator IRE-1 [51,52] and its target XBP-1 [52], which respond to an increase in unfolded proteins in the ER due to altered redox levels (Fig 2A). We used our assay to examine whether loss-of-function mutations in this core hypoxia-sensing pathway affected the ability of *C. elegans* to carry out RQ-dependent metabolism. If the hypoxia-sensing machinery is required for the switch from UQ-dependent aerobic metabolism to RQ-dependent anaerobic metabolism, mutants lacking the ability to sense hypoxia should be unable to survive 15 hours of KCN exposure.

We examined a set of 4 strains (see Table 1) containing mutations in the hypoxia sensing machinery and found no requirement for the core hypoxia-sensing pathway for *C. elegans* to survive long-term exposure to KCN (Fig 2B and S2 Fig). Worms homozygous for loss-of-function mutations in *hif-1* can survive 15 hours of KCN treatment as well as wild-type animals, as can strains with loss of function mutations in *ire-1* or *xbp-1*. Although *xbp-1* is significantly different ($1.00e-03 < p < = 1.00e-02$) than N2, there is only a ~25% reduction in recovery which is not sufficient for a potential drug target. This was a surprising result to us as we had expected that KCN treatment would activate a *hif-1*-dependent hypoxia response. To test whether KCN addition caused activation of HIF-1 target genes, and thus invoked a *hif-1*-dependent hypoxia response, we examined induction of expression of *nhr-57*, a well characterized transcriptional target of HIF-1 [53]. We find that doses of KCN up to ~50 μM result in potent induction of *nhr-57* expression as measured using a fluorescent transcriptional reporter (Fig 2C). Wild-type animals initially slow their movement when exposed to these lower doses of KCN, but subsequently recover full movement within ~3 hours; this acute recovery is *hif-1*-dependent (Fig 2D). However, at higher KCN doses (200 μM), we see no induction of *nhr-57* expression and no difference in movement between wild-type and *hif-1(ia4)* mutant animals —both strains slow their movement and become immobile after ~90 mins (Fig 2E). We also see no increase in *nhr-57* expression levels at higher KCN doses from RNA-seq data, suggesting the lack of GFP expression is unlikely to be due to quenching by KCN (S3 Fig). It thus appears that at lower KCN doses, worms invoke a *hif-1* transcriptional response that allows them to recover from acute KCN treatment, and this recovery is not RQ-dependent. At higher doses of KCN, worms do not activate *hif-1* but instead enter an immobile state where they rely on RQ-dependent metabolism. We suggest that the higher doses of KCN that are needed to drive worms to use RQ-dependent metabolism are mimicking anoxia rather than hypoxia. Consistent with this, *C. elegans* is known to be able to survive long periods of anoxia [54–58] and *hif-1* is not required for this survival [59].

Our data show that the switch to RQ-dependent metabolism in *C. elegans* does not require the core hypoxia-sensing pathway. Compounds that block the core hypoxia-sensing pathway are thus unlikely to affect the ability of STHs to activate RQ-dependent metabolism and this is a useful negative result since the hypoxia sensing machinery appeared an obvious candidate target for this. Biologically, it is also important: if *C. elegans* do not use the core hypoxia

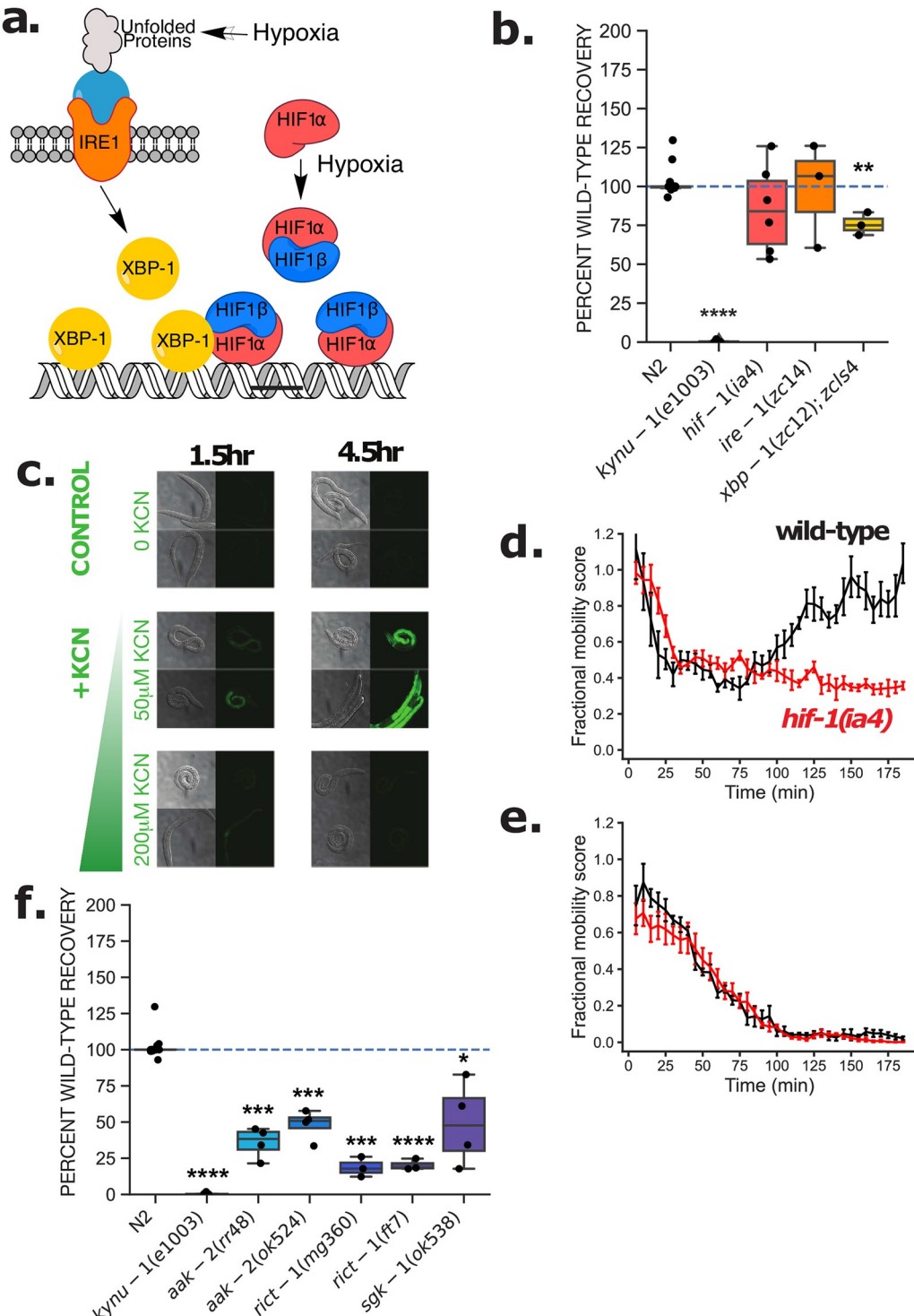

**Fig 2. The AMPK/mTOR pathway and not the hypoxia pathway is required for RQ-dependent metabolism.** (a) Model of the hypoxia response pathway. (b) Wild-type (N2) and mutant L1 worms for components of the hypoxia pathway were exposed to 200 μM KCN for 15 h. KCN was then diluted 6-fold and worm movement was measured over 3 h to track recovery from KCN exposure (see Materials and methods for details). *kynu-1(e1003)*, a mutant that lacks RQ and cannot survive prolonged KCN treatment was used as a positive control. Data points show biological replicates of movement scores of ~150 worms per well after 3 h recovery, with 2–3 technical replicates for each experiment. (c) Activation of the *nhr-57*::GFP reporter in response to KCN is dose-dependent. L1 worms were treated with 50 or 200 μM KCN and imaged after 1.5 and 4.5 hours. Worms incubated in M9 buffer alone were used as a negative control. Images are representative of 10–15 worms imaged. (d-e) Wild-type N2 (black curve) and *hif-1(ia4)* (red curve) L1 worms were

exposed to 50 (d) and 200 (e) μM KCN and their movement measured over 3 hours. Curves show the mean of 3 biological replicates. (f) *aak-2* has a 50–60% reduction in recovery (1.00e-04 < *p* < = 1.00e-03) and *rict-1* has a 75% reduction (1.00e-04 < *p* < = 1.00e-03). Wild-type (N2) and mutant L1 for components of the anoxia pathway were exposed to 200 μM KCN for 15 h. KCN was then diluted 6-fold and worm movement of ~150 worms per well was measured after 3 h to track recovery from KCN exposure (see Materials and methods for details). *aak-2(ok524)* is a loss-of-function mutant with a 408-bp deletion [32,33,60], while *aak-2(rr48)* is a semidominant-negative mutant with a substitution mutation [61]. Both *rict-1(ft7)* and *rict-1(mg360)* are loss-of-function mutants. *rict-1(ft7)* has a nonsense allele [62] while *rict-1(mg360)* has a missense allele [63].

pathway to induce the switch to RQ-dependent metabolism in hypoxic conditions, what is the switch? We address this in the next section.

## RQ-dependent metabolism is activated in response to metabolic consequences of anoxia

We found that the core hypoxia-sensing machinery is not involved in the switch to RQ-dependent metabolism. We reasoned that if *C. elegans* is not sensing levels of oxygen directly it might sense the consequences of reduced oxygen instead. Hypoxia has diverse effects on the organism including changes in metabolism [64]. One immediate consequence of the inability of *C. elegans* to use the aerobic mitochondrial ETC is a reduction in efficiency of ATP generation and thus a likely drop in ATP levels and growth rate. Reduced ATP levels are sensed by AMP-activated kinase [65], a heterotrimeric kinase which in *C. elegans* requires the *aak-1* and *aak-2* encoded catalytic subunits [61,66,67] and we also note that AMPK activity is required for *C. elegans* to survive anoxia [54]. We find that *C. elegans* requires AMPK activity to efficiently activate RQ-dependent metabolism (Fig 2F and S2 Fig) suggesting that the key switch is not a lack of oxygen per se but the metabolic consequences of reduced oxygen levels. We also examined whether the mTOR pathway is required for the shift to RQ-dependent metabolism. The mTOR complex is required for sensing whether there is sufficient nutrient availability to drive growth and is regulated by AMPK phosphorylation [68–70]. *rict-1* encodes the *C. elegans* orthologue of RICTOR [62,63], a core component of the mTOR complex and loss-of-function mutations in *rict-1* (Fig 2F and S2 Fig) prevent RQ-dependent recovery from KCN. We thus find that *C. elegans* also requires the mTOR complex for efficient activation of RQ-dependent metabolism. We conclude that *C. elegans* activates RQ-dependent metabolism via both AMPK and mTOR in response to dropping ATP levels in anoxic conditions.

Finally, we examined whether *sgk-1* is required for the switch to RQ-dependent metabolism in *C. elegans*. *sgk-1* encodes the serum and glucocorticoid induced kinase [63,71] and is involved in maintaining mitochondrial homeostasis [72], in mitochondrial stress responses [73], and it affects sensitivity to paraquat which generates Reactive Oxygen Species [74,75]. We found that worms homozygous for the *sgk-1(ok538)* loss-of-function mutation show a reduced (~50%) but variable (1.00e-02 < *p* < = 5.00e-02) recovery after 15 hours of KCN exposure which suggests it is also partially required for *C. elegans* to use RQ-dependent metabolism (Fig 2F and S2 Fig).

Together, our findings here indicate that the switch from UQ-dependent aerobic metabolism to RQ-dependent metabolism is not regulated by the core hypoxia-sensing machinery but instead requires the AMPK/mTOR machinery that senses nutrient levels including ATP levels and the SGK-1 kinase that responds to a variety of cellular stresses. AAK-2, RICT-1 and SGK-1 are all required for *C. elegans* to enter RQ-dependent metabolism and thus these three proteins are potential targets. We next examined each amino acid sequence to identify whether there were specific residues that are identical across the helminths studied (Fig 1A) and that

differ from those in a selection of their mammalian hosts (Fig 1A). Helminth species were selected so that 4/5 clades were represented and so that platyhelminthes were also included in the form of *Fasciola hepatica* (Fig 1A). We also included two molluscs and an annelid as their own group since they are also known to produce RQ [reviewed in 24]. Residues were considered of interest if they were conserved amongst the helminth (nematode+platyhelminth), *C. elegans* and mollusc+annelid group and were differentially conserved in mammalian hosts. We also checked whether any of those 'helminth-specific' residues reside at or near the active site.

AAK-1 and AAK-2 belong to a complex gene family whose evolutionary relationships were too complex for the method used here. In contrast, both RICT-1 and SGK-1 have clear evolutionary trees and conserved divergent residues between hosts and parasites. RICT-1 shows divergence between hosts and STHs in 22 residues. However, these are scattered throughout the protein and are not close to any binding interface or active site (S4 Fig). SGK-1 has 7 residues that differ between STHs and hosts and three that are near the active site. Two of these (residues 130 and 140 in *C. elegans)* are close to the active site while Phe136 is right at the active site [76,77]. This suggests it might be possible to generate an inhibitor that would block the STH SGK-1, disrupting the switch to RQ-dependent metabolism, without affecting the host orthologue. We note however that this active site change is not RQ-specific since it is present in species that do not make or use RQ (S5 Fig). These residues changes may or may not be sufficient to enable a helminth-specific SGK-1 drug. Taken together, while AMPK, mTOR, and SGK-1 all appear required for the efficient activation of RQ-dependent metabolism, only SGK-1 has STH-specific residues near the active site and these are not uniquely associated with RQ metabolism. We thus conclude that there is no high confidence target for anthelmintics within our candidate 'sensor' genes regulating the switch from UQ to RQ-dependent metabolism, though SGK-1 is perhaps still tractable.

## Multiple enzymes required for RQ synthesis have helminth-specific active sites

We and others previously showed that UQ and RQ synthesis share a similar pathway [34,35,78]. The critical difference in UQ and RQ synthesis is the precursor—UQ synthesis in animals uses 4-hydroxybenzoate (4HB) whereas RQ synthesis starts with 3-hydroxyanthranilate (3HA) as the precursor [34,78]. A crucial early step in synthesis of these distinct quinones is the prenylation of either of these two precursors by the polyprenyltransferase COQ-2—the substrate choice by COQ-2 is the key event that dictates whether UQ or RQ will be synthesized [35]. We recently found that the choice of substrate for COQ-2 is determined by a single alternative splicing event which changes the enzyme core from a conserved 4HB-utilizing enzyme to a helminth-specific 3HA-preferring enzyme [35]. The splice variant that uses 3HA as its preferred substrate has two helminth-specific residues deep in its active site making it an attractive drug target since it should be possible to identify inhibitors that specifically inhibit this RQ-synthesising form.

3HA is generated by metabolism of tryptophan via the kynurenine pathway (Fig 3A) [79,80]. In *C. elegans*, the kynurenine pathway requires the genes *tdo-2*, *kmo-1*, *amfd-1* and *kynu-1* and we previously showed that *kmo-1*, *amfd-1* and *kynu-1* are required for RQ synthesis [34]. These 3 enzymes are thus potential targets for anthelmintics—if we could identify a STH-specific inhibitor of any of these, it should block 3-HA synthesis and thus block RQ synthesis. Comparative genomics and structural modeling showed that the sole potential candidate is KMO-1 where there is a helminth-specific change in a critical region required for binding of kynurenine. This Tyr334Phe is not a major change so we do not believe it to be a

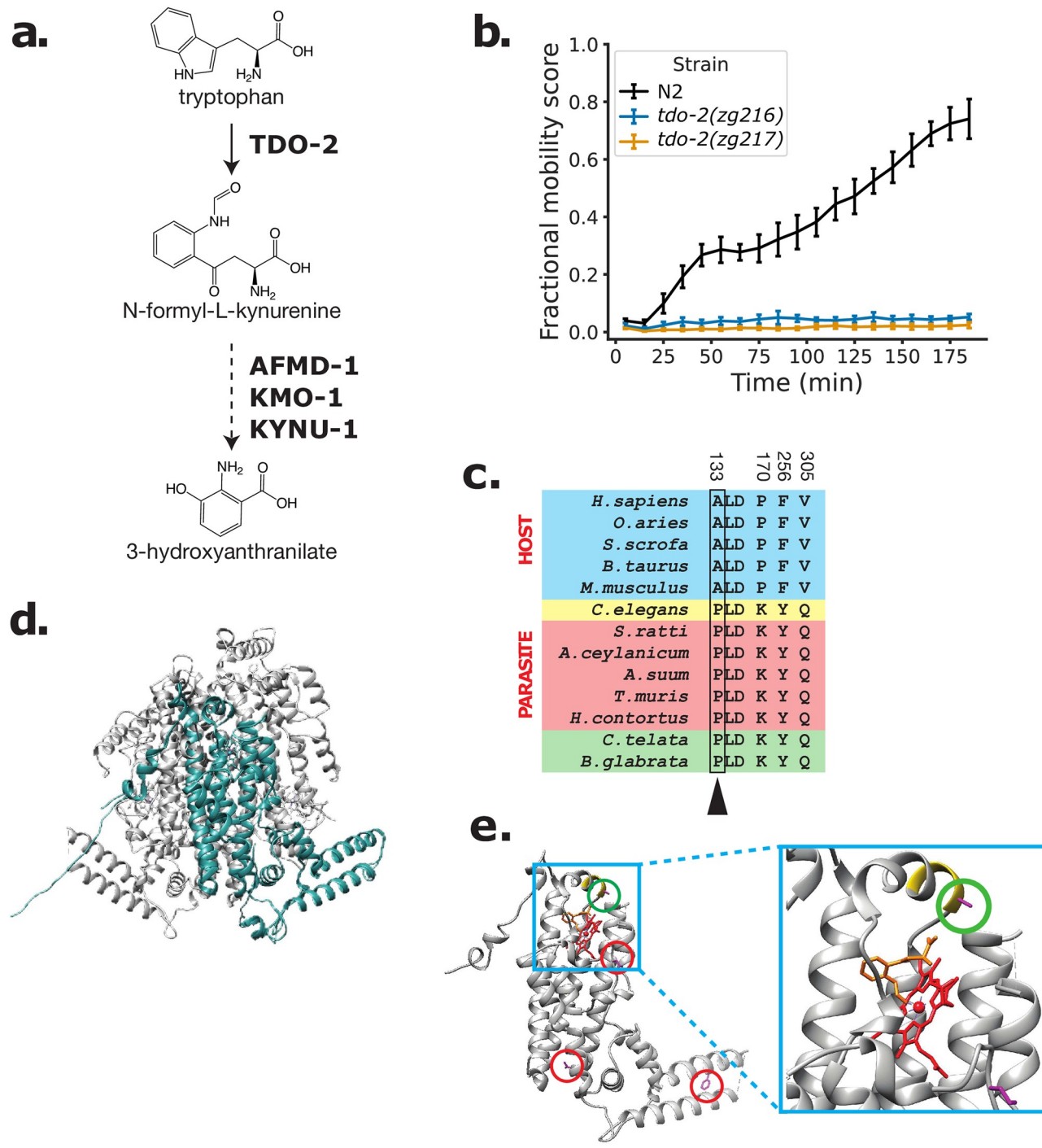

**Fig 3. TDO-2 is required for RQ-dependent metabolism.** (a) Schematic of kynurenine pathway. 3-hydroxyanthranilate (3HA), a precursor for RQ synthesis, is derived from tryptophan. Tryptophan 2,3-Dioxygenase (TDO; TDO-2 in *C. elegans*) is required for the oxidation of L-tryptophan to N-formyl-L-kynurenine while AMFD-1, KMO-1, and KYNU-1 function downstream and were previously shown to be required for RQ synthesis. (b) Wild-type and *tdo-2* loss-of-function mutants were treated with 200 μM KCN for 15 h followed by a 6-fold dilution with M9. Curves show the recovery from KCN treatment and represent the mean of 4 biological replicates and error bars are standard error. (c) Conserved divergent residues from host (blue), *C. elegans* (yellow), parasite (red) and mollusk/annelids (green) were identified using S3Det [38] with a cutoff of 2 (see Materials and methods). Residues are numbered according to *C. elegans*. One of the residues (arrow) is part of the PLD loop, a previously identified and evolutionarily conserved structural loop required for enzymatic activity [84]. (d) Crystal model of human TDO (PDB: 5TI9) with the single monomer highlighted in teal as visualized in Chimera [85]. (e) Single monomer of human TDO (PDB: 5TI9) in complex with Tryptophan and dioxygen with *C. elegans* sequence threaded over by homology (pink residues, circled in red). PLD loop is highlighted in yellow and the A vs P residue is circled in green and is close to tryptophan (orange) and to heme complexed with iron (red).

high priority target (S6 Fig). Our structural analyses did not find STH-specific residues at the active site of KYNU-1 (S7 Fig) while AMFD-1 protein sequences were too divergent from each other for our analysis pipeline. This suggests that it will be hard to identify STH-specific inhibitors to these 3 enzymes. We therefore turned our attention to Tryptophan Dioxygenase (TDO; TDO-2 in *C. elegans*) which catalyses the initial step in the kynurenine pathway [81–83], the oxidation of L-tryptophan to N-formyl-L-kynurenine.

We found that like *kmo-1*, *amfd-1* and *kynu-1*, *tdo-2* is also required for survival in conditions where RQ-dependent metabolism is required, confirming that it is also a potential target for anthelmintics (Fig 3B). We therefore examined TDO sequence and structure in STHs and their mammalian hosts. We identified 4 residues that consistently differ between STHs and their mammalian hosts (Fig 3C). One is of particular interest since it sits at the active site close to the haem and tryptophan binding site in a small loop (Fig 3D and 3E). This is the same region where characterized TDO inhibitors are known to bind [84,86,87] and which is required for the enzymatic activity of TDO-2 in *C. elegans* [84]. STHs and mammalian hosts thus differ in a single critical residue at the active site of TDO and we suggest that this single amino acid difference may make it possible to develop TDO inhibitors that will be helminth specific and that could block the generation of RQ precursors and thus block the synthesis of RQ itself.

Finally, we also examined COQ-3 and COQ-5, two genes downstream of COQ-2 in the pathway for RQ synthesis. Both these genes were previously shown to be required for RQ synthesis and thus are potential targets [78]. However, RNAi knockdown of *coq-3* and *coq-5* did not greatly affect the ability of the worm to carry out RQ-dependent metabolism (S8 Fig). COQ-5 has no obvious helminth-specific residues that differ consistently between mammalian host and helminth. COQ-3 has a single such residue but this is far from the active site (S9 Fig) and thus is unlikely to result in a helminth-specific inhibitor. We conclude that these are not good anthelmintic targets.

We have thus identified two excellent targets for potential anthelmintics in the RQ synthesis pathway: COQ-2 from our previous study [35] and TDO-2 as we describe here. Both are required for RQ synthesis in *C. elegans* and thus for RQ-dependent metabolism and both have helminth-specific residues at the active site.

## Two Quinone-coupled dehydrogenases are required for RQ-dependent metabolism

The direct consequence to energy metabolism of switching the main electron-carrying quinone from UQ to RQ is at the level of the quinone-coupled dehydrogenases (QDHs) that are connected to the mitochondrial electron transport chain (ETC) [24]. In most animals, these QDHs allow electrons to enter the ETC downstream from Complex I and onto the UQ pool to participate in oxidative phosphorylation. In helminths, however, these QDHs can act as either entry points or exit points for electrons. In aerobic conditions, they act as dehydrogenases where they shuttle electrons into the ETC by transferring electrons from substrates onto UQ, just as in other animals; however, in anaerobic conditions, they reverse direction and act as reductases where they transfer electrons from RQ and out of the ETC [23,26,28,88] (Fig 4A).

QDHs are excellent candidate targets for anthelmintics since they are the direct site of action of RQ. If we could selectively inhibit the QDHs that are required for RQ-dependent metabolism, we should kill the helminth parasite. STH QDHs have different quinone-binding requirements to those of their mammalian hosts: the only quinone that interacts with the host QDHs is UQ but QDHs in STHs and other RQ-using animals must permit both UQ and RQ to dock at their quinone-binding pockets [26]. In Complex II, this difference in quinone

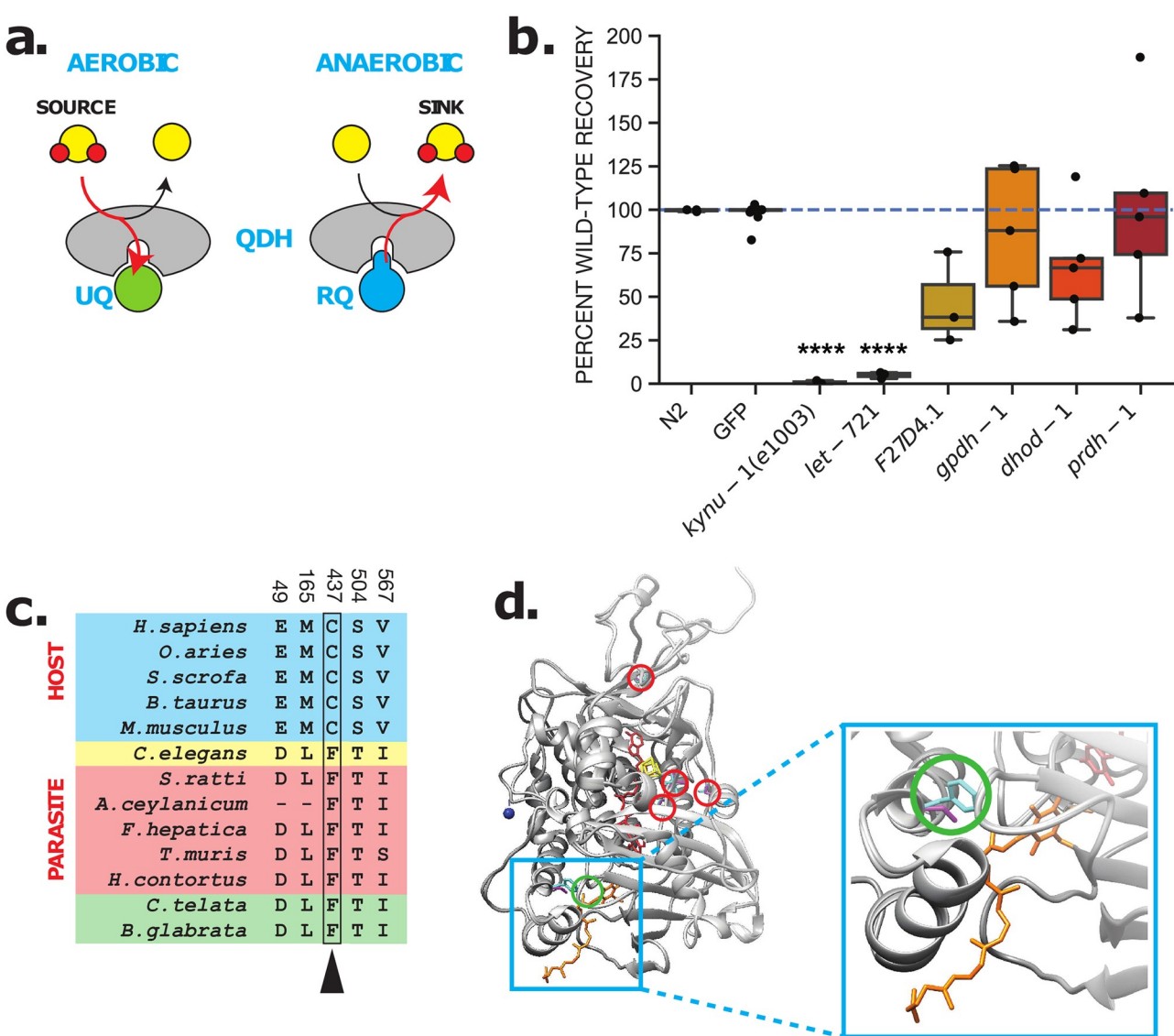

**Fig 4. ETFDH is required for RQ-dependent metabolism.** (a) Schematic showing role of QDHs in the electron transport chain in UQ and RQ-dependent metabolism. (b) Requirement for QDHs in RQ-dependent metabolism. RNAi knockdown of QDH genes was performed by feeding dsRNA-expressing bacterial strains to L1 worms until they developed to fertile adults. Their L1 progeny were subjected to 15 h of 200 μM KCN treatment followed by a 3 h recovery (see Materials and methods). An RNAi clone targeting GFP was used as a negative control. Data show movement scores after 3 h recovery and represent biological replicates, with 2–3 technical replicates in each. (c) Conserved divergent residues from host (blue), *C. elegans* (yellow), parasite (red) and mollusk/annelids (green) were identified using S3Det [38] with a cutoff of 2 (see Materials and methods). Residues are numbered according to *C. elegans*. Phe437 (arrow) is conserved in RQ-containing species but is not present in hosts. (d) Crystal structure of porcine ETFDH (PDB: 2GMH) in complex with UQ5 (orange) as visualized in Chimera [85] with *C. elegans* sequence (pink residues) and *H. sapiens* sequence (cyan residues) threaded over by homology (residues from (c) circled in red). The Phe437Cys residue (green) is located near the quinone binding site [89] where both RQ and UQ must be able to bind in *C. elegans* and parasites.

binding requirement between host and STH is known to have driven the divergence of the quinone-binding pocket and this has allowed the identification of Complex II inhibitors [31,90] that are highly selective for STH Complex II. These inhibitors include flutolanil [89] and wact-11 [31] and these inhibitors bind directly at the quinone-binding pocket [31,90]. We reasoned that if it is possible to generate inhibitors that selectively bind to the quinone-binding pocket

of Complex II, it may be possible to find inhibitors that selectively bind to the quinone binding pocket of other QDHs that use both UQ and RQ in helminths.

There are 5 QDHs in animals that are linked to the mitochondrial ETC. These comprise Succinate dehydrogenase (also known as Complex II) [26,91–93]; Electron-Transferring Flavoprotein (ETF) dehydrogenase [94–96]; Proline dehydrogenase [97–99]; Glycerol 3-phosphate dehydrogenase [100–102]; and Dihydroorotic acid dehydrogenase [103]. We note that there are other quinone-coupled enzymes in animals including sulphide:quinone oxidoreductase (SQRD-1 in *C. elegans* [104]) as well as orthologues of D- and L-hydroxyglutarate dehydrogenase [105–107] (D2HGDH and L2HGDH; F45D5.12 and Y45G12B.3 respectively). These are not thought to have any role in energy metabolism, our key focus here, and thus we have not pursued these in this study. We previously showed that Complex II is required for RQ-dependent metabolism in *C. elegans* and showed that helminth-specific residues in the quinone-binding pocket of Complex II in particular are important for RQ utilization[34]. However, we do not know which of the other QDHs may also be required and thus we examined the requirement for each QDH in our RQ-dependent metabolism assay.

We used RNAi to examine which of the QDHs are required for RQ-dependent metabolism using our RQ assay in *C. elegans*. We found that in addition to Complex II, ETFDH is also required for survival for 15 hours in KCN where worms rely on RQ-dependent metabolism (Fig 4B and S10 Fig). Complex II acts as a fumarate reductase in these conditions, transferring electrons from the RQ pool and generating succinate as a product. ETFDH links BCAA and fatty acid metabolism to the ETC during UQ-dependent aerobic metabolism [94,96,108,109] —when acting in reverse during RQ-dependent anaerobic metabolism, it transfers electrons from the RQ pool and generates BCAAs and short chain fatty acids including 2-methyl valerate and 2-methyl butanoate [95,110]. Since Complex II has been extensively studied as an anthelmintic target, we focused our attention on ETFDH.

We used sequence alignment to identify amino acids that consistently differ in ETFDH between hosts and helminths (Fig 4C) and find 5 such residues i.e. all helminths have one sequence, all mammlian hosts have a different sequence. We modelled *C. elegans* ETFDH (LET-721) on the *Sus scrofa* ETFDH (PDB:2GMH) crystal structure [89] to see where these helminth-specific amino acids sit. While some are dispersed around the structure, one of these host-helminth changes is located in the quinone-binding pocket of ETFDH (Fig 4D). It is a significant change: while mammalian hosts have a redox-sensitive cysteine at this position, helminths have the aromatic amino acid phenylalanine that may aid in charge delocalisation. This change right in the quinone-binding pocket suggests that just as it has been possible to identify potent helminth-specific succinate dehydrogenase inhibitors that bind to the quinone-binding pocket of Complex II, it should be possible to identify helminth-specific ETFDH inhibitors that bind to the quinone-binding pocket of ETFDH. We thus consider ETFDH to be another high confidence molecular target for anthelmintics.

## Metabolic enzymes that change directionality in RQ-dependent metabolism

One of the key features of the switch from UQ-dependent aerobic metabolism to RQ-dependent metabolism is the rewiring of a subset of core metabolic reactions [24,111,112]. This often takes the form of a reverse in the direction of a reaction. The canonical example is that of Complex II which uses succinate as a substrate and generates fumarate as a product during aerobic conditions; in anaerobic conditions, it uses fumarate as a substrate and generates succinate as a product [92,113]. There is a similar reverse in reaction direction for ETFDH and as we showed in this paper, ETFDH is also critical for RQ-dependent metabolism. This switch in

the direction of core reactions is not unique to quinone-coupled dehydrogenases and other helminth enzymes also switch their direction during RQ-dependent metabolism. The best characterized are those that involve the TCA cycle and specifically the reactions that lead to the generation of malate and fumarate and this is where we focus.

Malate metabolism is a key decision point between the aerobic and anaerobic TCA cycles in helminths. In aerobic conditions, malate is largely converted to oxaloacetate (OAA) by mitochondrial malate dehydrogenase (MDH-2 in *C. elegans*) as part of a normal animal TCA cycle [114]. In anaerobic conditions, however, malate is converted to fumarate by the enzyme fumarase (FUM-1 in *C. elegans*) acting in the reverse direction to the standard aerobic TCA cycle. Generation of fumarate is crucial since it acts as the substrate for RQ-coupled Complex II which is essential for RQ-dependent metabolism. Malate can be generated from a variety of substrates including from fumarate by fumarase (FUM-1), from OAA by reversal of malate dehydrogenases, from pyruvate either directly by the action of malic enzyme (MEN-1 in *C. elegans* [115]) or via OAA through the action of pyruvate decarboxylase (PYC-1 in *C. elegans* [115,116]). We used a combination of RNAi and classical genetics to test whether *fum-1*, *mdh-1*, *mdh-2*, *pyc-1* and *men-1* are required for RQ-dependent metabolism. In addition, we tested two genes, *F36A2.3* and *VF13D12L.3* that encode members of a distinct subfamily of malate dehydrogenases [117,118] that are typically found in archaeal genomes. Importantly for this study, these genes have no host homologues but are present in all STHs making them an attractive target.

As shown in Fig 5A, we find that none of the genes tested appear to be required for RQ-dependent metabolism in *C. elegans*. There are a number of caveats to this. First, *fum-1* and *mdh-2* are lethal by RNAi so we could not assess their involvement. This is not unexpected—they are required for the TCA cycle which is essential in both aerobic and anaerobic conditions. Second, *F36A2.3* and *VF13D12L.3* are closely related and the lack of a strong effect of these genes on RQ-dependent metabolism might be due to paralogue redundancy. We therefore used CRISPR genome engineering to make a presumptive null mutation in *VF13D12L.3* and carried out RNAi against *F36A2.3* in a strain homozygous for the *VF13D12L.3* null allele to reduce activity of both paralogues at once—we saw no effect on RQ-dependent metabolism (S11 Fig). This could reflect the lack of requirement for these genes, but could also be the result of poor knockdown of *F36A2.3* by RNAi. We suspect this to be the case since we used CRISPR engineering to make a presumptive null mutation in *F36A2.3* and found that this is lethal and we were unable to generate a strain without balancers. The fact that the mutant phenotype is lethal but the RNAi phenotype is not suggests that the *F36A2.3* RNAi is only giving a weak knockdown. It is intriguing to note that *F36A2.3* does not show any reproducible strong effect on RQ-dependent metabolism by RNAi but has a variable phenotype (1.00e-2 $< p <$ = 5.00e-2)—some biological repeats show strong effects, others very little. This is the sole gene for which we saw such variability and variable RNAi phenotypes can often result from weak or incomplete RNAi knockdown. We would thus not feel confident either excluding it as a possible target or including it as a high priority target based on this weak data. Overall, we conclude that we did not find that any of these metabolic enzymes to be definitely required for RQ-dependent metabolism in *C. elegans*.

More positively, the finding that *F36A2.3* is an essential gene in *C. elegans* suggests that this malate dehydrogenase is a good target for anthelmintics. All animals have both cytoplasmic and mitochondrial malate dehydrogenases (*mdh-1* and *mdh-2* in *C. elegans*). However, a small number of animal species have an additional family of malate dehydrogenases that is closely related to those found in archaea (Fig 5B). Intriguingly, no mammalian STH hosts have this third class of malate dehydrogenase but all species that make RQ (nematodes, STHs, annelids, and molluscs) have these. Crucially, since there is no similar gene in any STH mammalian

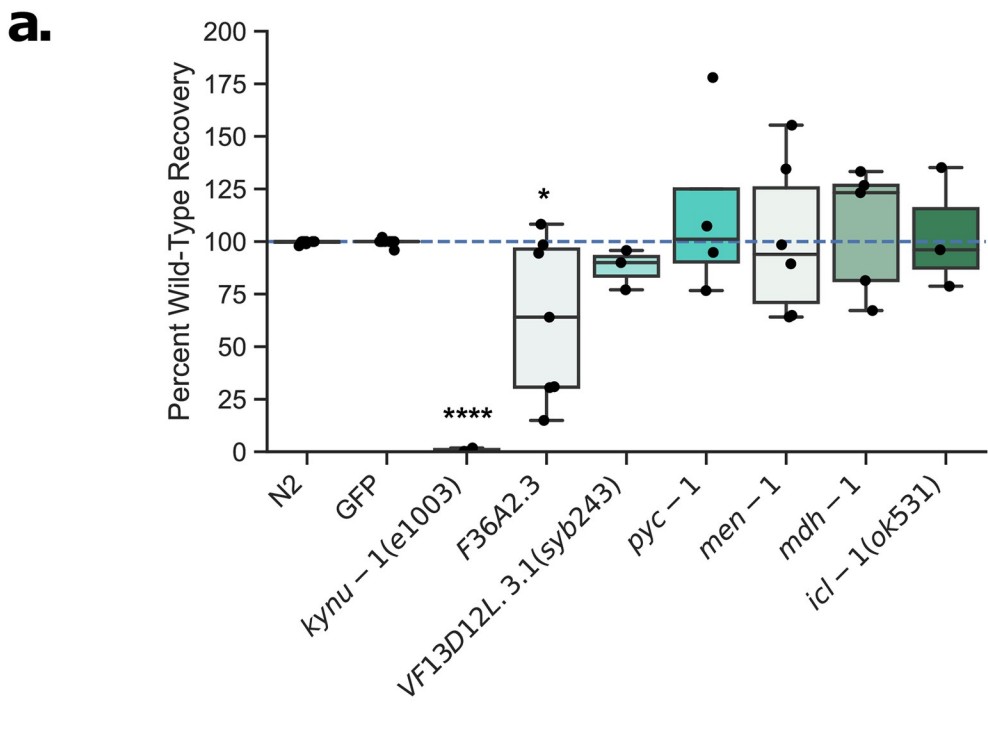

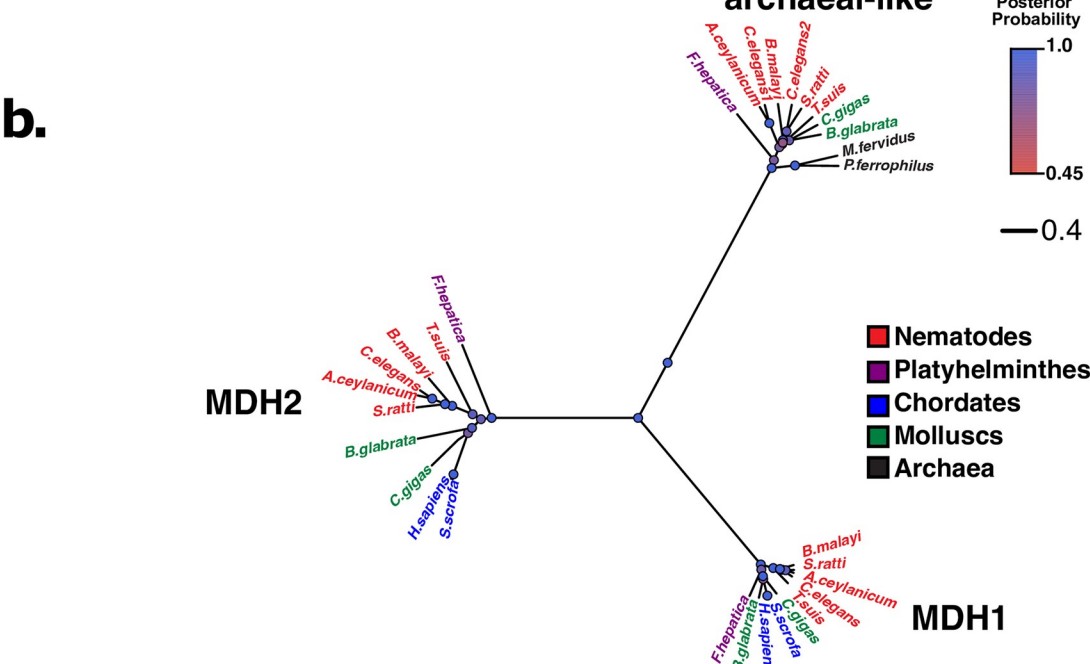

**Fig 5. Requirement for various metabolic enzymes for RQ-dependent metabolism.** (a) The involvement of various metabolic enzymes in RQ-dependent metabolism was tested using either loss-of-function mutants or RNAi knockdown. Wild-type, mutants, and RNAi-treated L1 worms were exposed to 200 μM KCN for 15 h followed by a 6-fold dilution, and worm movement was measured after 3 h recovery. For loss-of-function mutants, movement scores were compared to wild-type N2 worms, while for RNAi-treated worms, scores were compared to RNAi against a GFP control. Data show movement scores after 3 h recovery and represent biological replicates, with 2–3 technical replicates in each. (b) Evolutionary tree of the three types of malate dehydrogenase found in helminths as calculated by BEAST [119] (see Materials and methods) and visualized with FigTree. The colour of the circles reflects their posterior probability with the more blue being the most likely. Unlike hosts, helminths possess a third malate dehydrogenase which is more closely related to archaean MDHs (black).

host and since *F36A2.3* is an essential gene in *C. elegans*, it may be an additional target for new anthelmintics.

In addition to these changes in reaction direction, helminths can also truncate their TCA cycles to avoid the two steps where carbon atoms are lost in the form of $CO_2$. This may be beneficial during times of limited carbon source availability [120]. These steps occur in the serial conversion of isocitrate to alpha-ketoglutarate ($\alpha$-KG) and $\alpha$-KG to succinyl-CoA. The truncated TCA cycle that avoids these steps requires the isocitrate lyase ICL-1 to drive the glyoxylate shunt [121,122]. Plants, fungi and helminths can also carry out the glyoxylate shunt [123–125] but hosts have no ICL-1 homologues and thus cannot do this. ICL-1 has thus previously been proposed as a potential anthelmintic target [111,126]. We examined whether ICL-1 was required for RQ-dependent metabolism in *C. elegans*. As shown in Fig 5A, worms homozygous for a loss of function mutation in *icl-1* were no different to wild-type animals in their ability to carry out RQ-dependent metabolism and we conclude ICL-1 is not required at least within the assays used here.

Together, we conclude that while there are extensive metabolic changes downstream of quinone-coupled dehydrogenases that occur during RQ-dependent metabolism, none of the enzymes involved appear to be individually required for efficient RQ-dependent metabolism. They are thus not worthwhile targets for drugs designed to block RQ-dependent metabolism in STHs. We suggest that this may be because there are many parallel routes for the generation of the key metabolite malate including through multiple different malate dehydrogenases and malic enzyme and that inhibiting any single route has little effect. The sole target we identified here is *F36A2.3* which appears to be an essential gene rather than having a dedicated role in RQ-dependent metabolism. Since there is no mammalian homologue for this unusual malate dehydrogenase, it is potentially an excellent target for anthelmintics.

## Functional significance of STH-specific residues

We identified 4 enzymes that are required for RQ-dependent metabolism and that also have helminth-specific residues at their active sites: Complex II, ETFDH, COQ-2e and TDO-2. These are all high confidence targets for anthelmintics since it should be possible to identify compounds that inhibit the helminth and not the host enzyme and thus block RQ-dependent metabolism without affecting the mammalian host. We previously showed that mutations that affect the helminth-specific binding pocket of Complex II can block RQ-dependent metabolism and wanted to specifically test the functional significance of the helminth-specific residues in the other three targets we identified. We did this in two ways.

First, we looked at the phylogenetic distribution of the amino acids that differ between STHs and their mammalian hosts. These differences at the active sites could have arisen through random genetic drift and have no functional significance. Alternatively, the amino acid changes seen at the active sites of the STH enzymes may be important for their function in RQ-dependent metabolism—if so we would expect to see the same residues in the other animal lineages that use RQ, molluscs and annelids. If they are simply the result of genetic drift, we would not expect this distribution. We see identical amino acids in the active sites of COQ-2e, TDO-2, and ETFDH in STHs, annelids, and molluscs but not in any of the mammalian hosts of STHs. This suggests that these residues may be functionally important for the ability to do RQ-dependent metabolism.

Second, we used genetics to directly test whether the identified residues are critical for the ability to carry out RQ-dependent metabolism. For COQ-2, we had previously shown that all RQ-using animals make an unusual splice form (COQ-2e in *C. elegans*) that includes the STH-specific residues [34], but did not confirm the functional importance of these two amino acids

for RQ-synthesis. Here, we used CRISPR genome engineering to change these two residues in the COQ-2a isoform. COQ-2a has the pan-eukaryotic active site and only participates in UQ synthesis; we previously showed that animals that can only make the COQ-2a isoform cannot make RQ and thus cannot survive in conditions where RQ-dependent metabolism is required. However, here we generated a *C. elegans* strain in which animals cannot make the RQ-synthesising COQ-2e isoform but can only make a CRISPR-edited version of COQ-2a where two amino acids in COQ-2a are changed from the pan-eukaryotic consensus to the STH-specific residues in the COQ-2e splice form (Fig 6A). If these two residues are indeed sufficient for RQ synthesis, this strain that expresses only the edited COQ-2a form should now be able to make RQ and thus to survive in conditions where RQ is required. We find that mutant worms with this edited COQ-2a form can survive where RQ is required (Fig 6B), suggesting that this mutant can now make RQ. This demonstrates that these two STH-specific residues are indeed critical for RQ synthesis. In the case of TDO-2, we took advantage of an existing mutant, *tdo-2 (zg218)* [84]. The critical STH-specific amino acid in the active site, P133 in *C. elegans* TDO-2, is in a short 3 amino acid loop (PLD, residues 133–135 in *C. elegans*). Crucially, the mutant strain *tdo-2(zg218)* has been engineered to delete only that PLD loop [84]. We find that *tdo-2 (zg218)* animals with the deleted PLD loop cannot carry out RQ-dependent metabolism confirming the functional importance of this STH-specific region (Fig 6C). Finally for ETFDH, we also used CRISPR genome engineering to switch the single STH-specific amino acid in the quinone binding pocket to the mammalian host consensus. We find that this single change reduces the ability of *C. elegans* to carry out RQ-dependent metabolism, confirming its functional importance, though we note the effect is weaker (Fig 6D).

Taken together, both the phylogenetic distribution of the active site amino acid changes and the direct functional testing show that not only do STHs have changes at the active sites of COQ-2e, TDO-2, and ETFDH, compared to their hosts but these active site changes are critical for their ability to use RQ-dependent metabolism. These three enzymes are thus high confidence targets for new anthelmintics.

## Discussion

Soil transmitted helminths (STHs) are major human pathogens and anthelmintics like ivermectin [5,6], benzimidazoles [8], and levamisole [7] are listed by the WHO as essential medicines [9]. These anthelmintics are powerful but there is a limited number of drug classes that are widely available and resistance is rising to these in both human and animal populations. New classes of anthelmintics are thus needed to combat these pathogens.

There are two fundamental strategies to identify potential new anthelmintics. The first is unbiased screening for compounds that can kill helminths but that do not affect their mammalian hosts. This type of unbiased screening has been extremely successful and is the source of all major classes of anthelmintics to date. The screens for these compounds were not directed at any specific helminth target proteins and targets were elucidated afterwards largely through genetic screens. The second, alternative approach would be to search for compounds that inhibit a pre-defined helminth target protein without affecting the mammalian host orthologues. Such target-directed approaches allow powerful, *in silico* docking methods to identify candidate inhibitors and such strategies have been successful in identifying drugs that specifically inhibit individual kinases that are mutated in cancers, for example [127,128] However, no such target-directed approaches have been successful thus far for the identification of anthelmintics [129]. One reason may be the lack of validated targets. Our goal in this study is to provide a set of validated high confidence helminth targets to allow *in silico* methods for the selection of new anthelmintics.

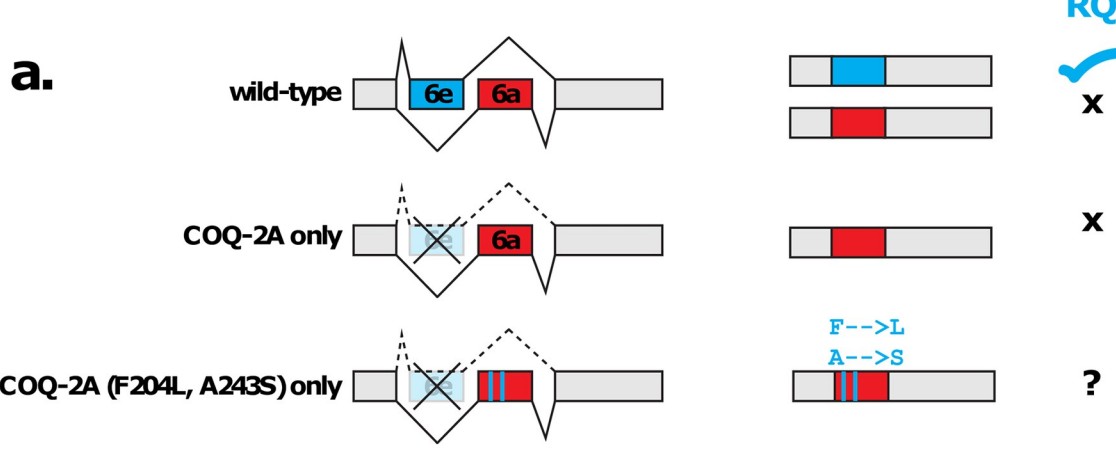

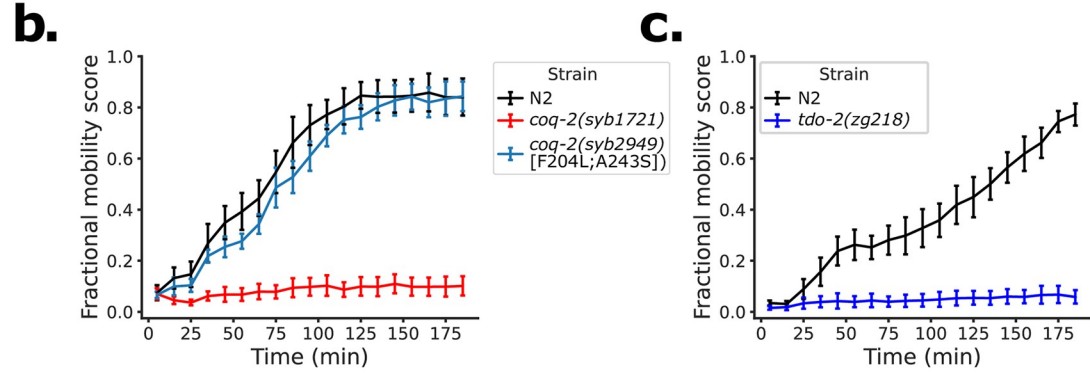

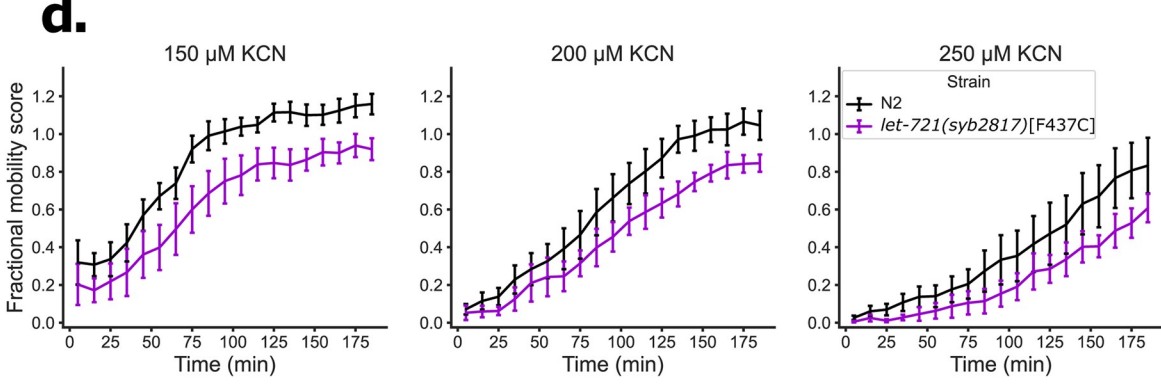

**Fig 6. Mutations in STH-specific residues affect ability to survive extended KCN treatment.** (a) Schematic of CRISPR-mediated engineering of a COQ-2 mutant. *coq-2* exons 6a and 6e are alternatively spliced in a mutually exclusive manner in wild-type worms. Mutant worms with exon 6e deleted (*coq-2Δ6e*) are unable to synthesise RQ. Exon 6e contains 2 helminth-specific residues and mutations were generated in the *coq-2Δ6e* mutant, such that two residues in exon 6a are replaced with their helminth-specific counterparts (F204L, A243S). (b) Helminth-specific residues in COQ-2 are required for survival in KCN. L1 worms were treated with 200 μM KCN for 15 h followed by a 6-fold dilution, and worm movement was tracked for 3 h. The RQ-deficient *coq-2Δ6e* mutant cannot survive extended KCN treatment while the *coq-2Δ6e* F204L;A243S mutant (*coq-2a(syb2949);coq-2e(syb1721)*) can now recover from KCN treatment, suggesting restored ability to synthesise RQ. (c) The TDO-2 PLD loop is required for survival in KCN. Wild-type (N2, black) or *tdo-2(zg218)* mutant worms (blue) L1 worms were treated with 200 μM KCN for 15 h followed by a 6-fold dilution, and worm movement was tracked for 3 h

(d) LET-721 helmint-specific quinone binding pocket is required for normal survival in KCN. Wild-type (N2, black) or *let-721* mutant L1 worms were treated with 150–250 μM KCN for 15 h followed by a 6-fold dilution, and worm movement was tracked for 3 h. All curves show the mean of 4 biological replicates and error bars represent standard error.

The critical issue for any *in silico* approach to identify potential new anthelmintics is the selection of the target. Helminth genomes typically encode over ~20,000 genes. Which of these is the best target for a potential anthelmintic? We drew insight in target selection from known anthelmintic targets. Unexpectedly, while helminth genomes contain many helminth-specific genes and gene families [130,131], few of these has emerged as a target of major anthelmintics to date and anthelmintics typically target proteins that are shared between parasites and their mammalian hosts. We note ivermectin and monepantel as exceptions to this—ivermectin targets glutamate-gated chloride ion channels that are not present in vertebrates [5,6] while monepantel targets the nematode-specific DEG-3 subfamily of acetylcholine receptors [132]. However, in most cases, the selectivity of anthelmintics for helminths is due to subtle differences between the helminth and host protein sequences. For example, both helminths and hosts have nicotinic acetylcholine receptors (nAChRs) that are critical for signalling within the nervous system [133–135]. Levamisole acts as a powerful anthelmintic by binding a specific subclass of nAChR that is only found in helminths [136,137], hence its selectivity for paralysing the parasite while leaving the host unaffected. Benzimidazoles target another highly conserved target, beta-tubulin [138,139]. However, subtle sequence changes between host and helminth beta-tubulin results in high selectivity of benzimidazoles for helminths [140–142]. These studies indicate that an excellent source of potential anthelmintics targets are proteins that are conserved between host and parasite, but where small sequence differences allow the selection of compounds that only bind the parasite protein with high affinity. These are the types of targets we pursue here.

To identify potential anthelmintic targets, we focused on the difference in energy metabolism between parasites and their mammalian hosts and how these have shaped the active sites of shared enzymes. Because mammalian hosts only make and use UQ whereas STHs must make and use both RQ and UQ to switch to RQ-dependent anaerobic metabolism in the low oxygen niches of their hosts, several of these enzymes have undergone critical residue changes that has allowed us to identify key targets for potential new classes of anthelmintics. The best characterised molecular change in STHs that is driven by the requirement to be able to use both UQ and RQ is that in the quinone binding pocket of Complex II, succinate dehydrogenase [20,90,92,113,143,144]. Amongst the multiple amino acid changes around the quinone-binding pocket of Complex II, we have previously showed that single residue mutations in this pocket can alter the ability of *C. elegans* to use RQ-dependent metabolism suggesting that these helminth-specific residues are likely critical for the docking of RQ [34]. The helminth-specific Complex II inhibitors flutolanil [145] and the related compound wact-11 [31] both rely on the differences in the quinone binding pocket between STH and host Complex II. This requirement of the enzyme for RQ-dependent metabolism and the STH-specific changes to the quinone binding pocket have made it an excellent anthelmintic target. We therefore examined other quinone-coupled dehydrogenases and identified ETF dehydrogenase (ETFDH) as also being required for RQ-dependent metabolism in *C. elegans*. Excitingly, we found that STHs have a single amino acid change in the quinone-binding pocket of ETFDH relative to the host sequences suggesting that like Complex II, ETFDH may also be an excellent target for anthelmintics.

To expand our list of potential targets beyond quinone-coupled enzymes, we tested a wider set of candidate genes that all might have potential roles in RQ-dependent metabolism and thus might be targets for anthelmintics. These included genes that regulate the switching from

UQ synthesis to RQ synthesis, the genes required for RQ synthesis, and the metabolic enzymes downstream of quinone-binding enzymes such as those involved in malate metabolism and the TCA cycle.

In total, we identified 4 such high priority targets. Two, TDO-2 and COQ-2e, act in the RQ synthesis pathway and the other two, Complex II and ETFDH, are the direct docking sites for RQ where it can transfer electrons out of the ETC and onto terminal acceptors. All four targets are required for RQ-dependent metabolism and all four also have specific changes between STHs and their hosts in critical regions of the proteins that may allow the development of STH-specific inhibitors. If the specific changes we identify in these proteins are important for the ability to make and use RQ, we would expect similar or identical changes in the other animal lineages that make RQ, molluscs and annelids. We find that molluscs and annelids have the exact same amino acid changes in TDO-2, COQ-2e, and ETFDH as we find for STHs. While this could be the result of identical random drift in all three lineages, it is more likely related to their use of RQ as a key electron carrier.

Crucially, to directly test the functional importance of these active site changes, we used genome engineering to explicitly test the importance of each of the identified residues for RQ synthesis. We find that mutating the active sites of COQ-2, TDO-2 or Complex II from their STH-specific form to the host version abolished the ability of *C. elegans* to do RQ-dependent metabolism and mutating the single STH-specific residue in the quinone-binding pocket of ETFDH to the host form reduced RQ-dependent metabolism. Taken together this shows that the residues that differ between STHs and their hosts in the active sites of COQ-2, TDO-2, ETFDH and Complex II are functionally critical for the ability of *C. elegans* to make and use RQ and we suggest that this should also be true for STHs. More importantly, the differences in active sites between STHs and their mammalian hosts in these critical enzymes should make it possible to use *in silico* methods to identify STH-specific inhibitors for these key enzymes. These may ultimately lead to new classes of anthelmintics which are urgently needed as resistance to current anthelmintics continues to rise.

## Materials and methods

### *Caenorhabditis elegans* strains and maintenance

The *F36A2.3 (syb769)/hT2*, *VF13D12L.3.1(syb243)*, *coq-2a(syb2949); coq-2e(syb1721)* and *let-721(syb2817)* mutant strains were generated by SunyBiotech (Fuzhou City, China) using the CRISPR-Cas9 system. The specific mutations were confirmed by DNA sequencing and details of each of these strains are provided in S1 Table. The *coq-2e(syb1721)* strain used was previously described [35] and was a kind gift from Prof. Gustavo Salinas. The *tdo-2(zg216)*, *tdo-2 (zg217)* and *tdo-2(zg218)* strains were a kind gift from Prof. Ellen Nollen [84]. The wild-type N2 strain and all other strains listed in Table 1 were obtained from the Caenorhabditis Genetics Center (University of Minnesota, USA), which is funded by NIH Office of Research Infrastructure Programs (P40 OD010440). All worm stocks were maintained at 20°C on NGM agar plates seeded with the *E. coli* strain OP50 as described elsewhere [146].

### Image-based KCN assays

The 15 h KCN recovery assay was performed as previously described [34,37]. In outline, L1 worms were washed from NGM agar plates and isolated by filtration through an 11 μm nylon mesh filter (Millipore: S5EJ008M04). Approximately 150 L1 worms in M9 were dispensed to each well of a 96 well plate and an equal volume of potassium cyanide (KCN) (Sigma-Aldrich St. Louis, MO) solution was then added to a final concentration of 150–250 μM KCN. Upon KCN addition, plates were immediately sealed using an aluminium seal and incubated at room

temperature for 15 h on a rocking platform. After 15 h, the KCN was diluted 6-fold by addition of M9 buffer. Plates were then imaged on a Nikon Ti Eclipse microscope every 10 min for 3 h.

After imaging, fractional mobility scores (FMS) were calculated using a custom image analysis pipeline [37]. For each strain, FMS scores for the KCN-treated wells were normalized to the M9-only control wells for the WT strain at the first timepoint. FMS scores of the non-WT wells were then normalized for each timepoint to the average of N2 (mutant) or HT115 treated with GFP RNAi (RNAi) wells that were also treated with 200μM KCN. For endpoint data, an average of the normalized FMS scores at 160, 170 and 180 min was taken for each biological replicate and 2–3 technical replicates were carried out for each experiment. Welsh's *t*-test was used to test for the statistical significance between WT/GFP RNAi and the mutants/RNAi.

For the acute 3 h KCN assay, L1s were isolated and treated with KCN as described above. After KCN addition, the plates were immediately imaged every 5 min for 3 h. FMS scores of KCN-treated wells were then normalized to the M9-only control wells at each timepoint.

## Analysis of the *nhr-57*::GFP reporter

The *nhr-57*::GFP reporter [53] was used to assay HIF-1 activity after KCN treatment. L1 worms were purified by filtration and treated with 40 μl KCN of various concentrations. Worms incubated in M9 alone were used as the control. After 1.5 h and 4.5 h of KCN treatment, worms were imaged using a Leica DMRE fluorescence compound microscope, with levamisole used as an anesthetic. All images were taken using the 40x objective.

## RNAi experiments

RNAi-by-feeding was performed as previously described [146]. RNAi clones were grown overnight at 37˚C in LB media with 1 mM carbenicillin and seeded onto NGM plates supplemented with 1 mM IPTG and 1 mM carbenicillin. For each RNAi knockdown, approximately 350 wild-type L1 worms were grown on NGM plates seeded with dsRNA-expressing bacteria. GFP-targeting dsRNA-expressing bacteria was used as a negative control. After one generation, worms were washed with M9 buffer and filtered to collect L1 progeny. These worms were then transferred to 96-well plates to be used in the KCN recovery assay as described above.

## Sequence identification

To identify host, parasite, mollusk/annelid and invertebrate sequences, four search strategies were made using BLASTP (protein databases). The search strategies are detailed below. For each search, the first 100 sequences which met at minimum 30% query cover were selected for further processing. Sequences were retrieved from NCBI protein and nucleotide databases and from RCSB Protein Data Bank (PDB).

For parasites, each search was made in the non-redundant protein sequences (nr) database against *Fasciola hepatica* (taxid:6192), *Ancylostoma ceylanicum* (taxid:53326), *Strongyloides ratti* (taxid:34506), *Brugia malayi* (taxid:6279), *Trichuris suis* (taxid:68888), *Haemonchus contortus* (taxid:6289) and *Ascaris suum* (taxid:6253). For hosts as well as *C. elegans*, each search was made in the reference proteins (refseq_protein) database against *Caenorhabditis elegans* (taxid:6239), *Ovis aries* (taxid:9940), *Bos taurus* (taxid:9913), *Mus musculus* (taxid:10090), *Sus scrofa* (taxid:9823) and *Homo sapiens* (taxid:9606). For mollusks and annelids, each search was made in the nr database against *Magallana gigas* (taxid:29159), *Biomphalaria glabrata* (taxid:6526) and *Capitella telata* (taxid:283909). In order to capture a broad select of other invertebrates, the search strategy used the refseq_protein database included *Insecta* (taxid:50557), *Arachnida* (taxid:6854), *Mollusca* (taxid:6447), *Crustacea* (taxid:6657), *Anthozoa* (taxid:6101), *Onychophora* (taxid:27563), *Xiphosura* (taxid:6845), *Cnidaria* (taxid:6073), *Echinodermata* (taxid:7586) and

*Porifera* (taxid:6040) while excluding *Vertebrata* (taxid:7742), *Crassostrea* (taxid:6564), *Biomphalaria* (taxid:6525), *Capitella* (taxid:51293) and *Drosophilidae* (taxid:7214).

The crystal structures were found on PDB by searching for the name of the enzyme and finding the best match, preferentially in a host or parasite and/or one bound with the ligand(s) or an inhibitor.

### Sequence analysis

Sequences were aligned using Clustal Omega [147] with {msa} [148] then ModelTest-NG [149] was used to identify the substitution model and its parameters for BEAST [119]. Speciation was based on the Yule Process [150,151] and a random local clock was used. BEAST was run using the CIPRES Science Gateway [152]. Tracer was used to confirm that an effective sample size (ESS) had been reached [153] and the trees were summarized using TreeAnnotator with a burnin percentage of 10% and a posterior probability limit of 0.5. Trees were visualized using FigTree.

Based on the distribution of sequences in the tree, sequences were selected for each species. Two sets of sequences were selected for each protein; one included only host, mollusk/annelid, *C. elegans* and parasite sequences in addition to any further crystal structure sequences while the second also included the expanded invertebrate sequences. If a sequence was clustered with other sequences from the same species, only one sequence was selected with a preference for first PDB sequences and then XP sequences. If there were multiple clear gene families across species, one sequence per species was selected based on gene families. Gene families which included sequences from other proteins outside of the protein of interest were not included.

Selected sequences were realigned using Clustal Omega [147] and {msa} [148] then visualized in JDet [154]. S3det [38] was used with a cut-off of 2 with categories based on host, mollusk/annelid, *C. elegans* and parasite. In cases where there had been a gene family expansion, additional groups were used. These residues were also checked against the expanded invertebrate list in order to determine whether they were RQ-specific.

### Protein analysis

Crystal structures were visualized using Chimera [85] and the *C. elegans* and *H. sapiens* (where needed) sequences were threaded by homology using Modeller [155,156]. Active sites were determined based on the original published crystal structures. Residues of interest were visualized relative to ligands and known active sites.

### Supporting information

**S1 Fig. 15 h KCN treatment drives *C. elegans* into using RQ-dependent metabolism at different developmental stages.** (a) RQ is required for survival following extended treatment with KCN at all *C. elegans* larval stages of development. Wild-type (N2, red boxes) and *kynu-1 (e1003)* mutant animals (blue boxes) at various stages were exposed to 200 μM KCN for 15 h. KCN was then diluted and worm movement was measured after a 3 h recovery. Boxplots show the worm mobility scores 3 h after recovery from KCN and comprise 3–4 independent biological replicates. Significance of difference between N2 and *kynu-1(e1003)* was calculated using a t-test (** = $p < 0.01$). (b) Movement of N2 (red curve) and *kynu-1(e1003)* (blue curve) worms for the 3 h period after recovery from 200 μM KCN. Curves show the mean of 3–4 biological replicates and error bars represent standard error.
(TIF)

**S2 Fig. Movement of various *C. elegans* mutants after recovery from 15 h KCN treatment at various KCN concentrations.** Wild-type (N2, grey curve) and mutant worms (blue and purple curves) defective in components of the hypoxia pathway were exposed to 150–500 μM KCN for 15 h. KCN was then diluted 6-fold and worm movement was measured over 3 h to track recovery from KCN exposure.
(TIF)

**S3 Fig. *nhr-57* expression does not increase after treatment with 200 μM KCN.** Expression levels (RPKM) of *nhr-57* from RNA-seq data. L1 *C. elegans* worms were treated with 200 μM KCN and samples were collected 0, 3 and 15 hours after incubation with KCN.
(TIF)

**S4 Fig. Conserved divergent residues are dispersed throughout RICT-1.** (a) RICT-1 is a component of the mTOR2 complex which regulates a number of downstream processes including SGK-1. (b) Conserved divergent residues of RICT-1 with a cutoff of 2 using S3Det [38] shows conserved divergence though much of it is not RQ-specific. (c) Crystal structure of human mTORC2 (PDB: 5ZCS) as visualized in Chimera [85] with RICT-1 in green. *C. elegans* sequence (pink residues) are threaded over by homology but are dispersed throughout the protein and seem more likely to affect its dynamics than be druggable.
(TIF)

**S5 Fig. Conserved divergence in SGK-1 is not RQ-specific.** (a) Conserved divergent residues of SGK-1 with a cutoff of 2 using S3Det [38] shows conserved divergence though much of it is not RQ-specific but rather reflects the SGK1/2/3 variation within hosts. The same is true of the previous characterized active site [76,77]. (b) Crystal structure of human SGK-1 (PDB: 3HDM/2R5T) as visualized in Chimera [85]. *C. elegans* sequence (pink residues) are threaded over by homology.
(TIF)

**S6 Fig. A single conserved divergent residue is within the active site of KMO-1.** (a) KMO-1 catalyzes a key reaction in the biosynthetic pathway which makes the 3HA, a RQ precursor. (b) Conserved divergent residues of KMO-1 with a cutoff of 2 using S3Det [38] shows poor conservation amongst helminths while the known active site [157] does have a conserved divergent residue (green arrow). (c) Crystal structure of *P.fluorescens* KMO-1 (PDB: 5NAK) and *H.sapiens* KMO-1 (PDB: 5X68) as visualized in Chimera [85] with *C. elegans* sequence (pink residues) threaded over by homology. A single conserved divergent residue is within the substrate binding active site (blue).
(TIF)

**S7 Fig. No conserved divergent residues are near KYNU-1's active site.** (a) KYNU-1 catalyzes a key reaction in the biosynthetic pathway which makes the 3HA, a RQ precursor. (b) Conserved divergent residues of KYNU-1 with a cutoff of 2 using S3Det [38] shows strong conserved divergence while the known active site [158] does not. (c) Crystal structure of human KYNU-1 (PDB: 3E9K) as visualized in Chimera [85] with *C. elegans* sequence (pink residues) threaded over by homology.
(TIF)

**S8 Fig. Effects of RNAi knockdown of *coq-3* and *coq-5* on worm recovery after extended KCN treatment.** RNAi knockdown of *coq-3* and *coq-5* was performed in L1 worms and after one generation their L1 progeny were subjected to 15 h 200 μM KCN treatment followed by a 3 h recovery (blue curve, see Materials and methods). An RNAi clone targeting GFP (grey

curve) was used as a negative control. Curves show the mean of 3 biological replicates and errors bars represent standard error.
(TIF)

**S9 Fig. Conserved divergent residues of COQ-3 are not close to the active site.** (a) COQ-3 catalyzes several reactions in the UQ/RQ biosynthetic pathway. (b) Conserved divergent residues of COQ-3 with a cutoff of 2 using S3Det [38] shows poor conservation amongst helminths while the known active site [159] is either conserved or very flexible in helminths as with 211. (c) Crystal structure of *E.coli* COQ-3 (PDB: 4KDC) as visualized in Chimera [85] with *C. elegans* sequence (pink residues) and *H.sapiens* sequence (cyan residues) threaded over by homology. Conserved divergent residues are not near the active site (blue).
(TIF)

**S10 Fig. RNAi knockdown of *let-721* results in reduced recovery from extended KCN treatment.** (a) RNAi knockdown of the ETF subunit alpha gene *F27D4.1* (yellow curve) and the ETFDH gene *let-721* (blue curve) was followed by a 15 h KCN treatment at various concentrations and recovery of movement was tracked for 3 h (see Materials and methods). An RNAi clone targeting GFP (grey curve) was used as a negative control. (b) RNAi knockdown of various other QDH genes (blue curve) was performed as described above and worms were subjected to 15 h 200 µM KCN treatment. An RNAi clone targeting GFP (grey curve) was used as a negative control. All curves show the mean of at least 3 biological replicates and error bars represent standard error.
(TIF)

**S11 Fig. RNAi knockdown of *F36A2.3* in a *VF13D12L.3* loss-of-function mutant does not affect recovery from KCN treatment.** (a) Effect of loss of archaeal MDH genes on the ability of *C. elegans* worms to survive extended KCN exposure. RNAi was performed in either wild-type or *VF13D12L.3.1(syb243)* L1 worms and their progeny were subjected to 15 h of 200 µM KCN treatment followed by a 3 h recovery from KCN treatment (see Materials and methods). RNAi targeting GFP was used as the negative control. Boxplots show the mobility scores of 4 independent biological replicates after 3 h recovery period and are normalised relative to the GFP control in wild-type N2 worms. The RNAi and mutant combination shows no significant difference in recovery using a two-way ANOVA test ($p = 0.52$). (b) Worm movement during the 3 h recovery period from KCN. Curves show the mean of 4 biological replicates and error bars represent standard error.
(TIF)

**S1 Table. *C. elegans* strains generated using the CRISPR-Cas9 system.**
(XLSX)

**S2 Table. Previous researchers have concluded that several metabolic genes are key for RQ-dependent metabolism.**
(XLSX)

**S1 Data. Data for Fig 2b.**
(CSV)

**S2 Data. Data for Fig 2f.**
(CSV)

**S3 Data. Data for Fig 3b.**
(TXT)

**S4 Data. Data for Fig 3c–3e.**
(FASTA)

**S5 Data. Data for Fig 4b.**
(CSV)

**S6 Data. Data for Fig 4c and 4d.**
(FASTA)

**S7 Data. Data for Fig 5a.**
(CSV)

**S8 Data. Data for Fig 5b.**
(TXT)

**S9 Data. Data for Fig 6b.**
(TXT)

**S10 Data. Data for Fig 6c.**
(TXT)

**S11 Data. Data for Fig 6d.**
(CSV)

**S12 Data. Data for S1 Fig.**
(TSV)

**S13 Data. Data for S2 Fig.**
(TSV)

**S14 Data. Data for S3 Fig.**
(TSV)

**S15 Data. Data for S4 Fig.**
(FASTA)

**S16 Data. Data for S5 Fig.**
(FASTA)

**S17 Data. Data for S6 Fig.**
(FASTA)

**S18 Data. Data for S7 Fig.**
(FASTA)

**S19 Data. Data for S8 Fig.**
(TXT)

**S20 Data. Data for S9 Fig.**
(FASTA)

**S21 Data. Data for S10 Fig.**
(TXT)

**S22 Data. Data for S11 Fig.**
(TSV)

## Acknowledgments

We thank CGC for multiple strains and Prof. Ellen Nollen for the kind gift of the *tdo-2* strains. We thank Dr Amy Caudy and Prof. Peter Roy for advice and helpful discussions.

## Author Contributions

**Conceptualization:** Margot J. Lautens, June H. Tan, Andrew G. Fraser.

**Data curation:** Margot J. Lautens, June H. Tan.

**Formal analysis:** Margot J. Lautens, June H. Tan, Xènia Serrat, Samantha Del Borrello.

**Funding acquisition:** Andrew G. Fraser.

**Investigation:** Margot J. Lautens, June H. Tan, Xènia Serrat, Samantha Del Borrello, Michael R. Schertzberg.

**Methodology:** Margot J. Lautens, June H. Tan, Xènia Serrat, Michael R. Schertzberg.

**Project administration:** Andrew G. Fraser.

**Software:** Margot J. Lautens, June H. Tan.

**Supervision:** Andrew G. Fraser.

**Validation:** Margot J. Lautens, June H. Tan.

**Visualization:** Margot J. Lautens, June H. Tan, Xènia Serrat, Samantha Del Borrello, Andrew G. Fraser.

**Writing – original draft:** Andrew G. Fraser.

**Writing – review & editing:** Margot J. Lautens, June H. Tan, Xènia Serrat, Andrew G. Fraser.

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
