## [Decision Letter · Decision Letter 0]

25 Aug 2021

Dear Prof. Fraser,

Thank you very much for submitting your manuscript "Identification of enzymes that have helminth-specific active sites and are required for Rhodoquinone-dependent metabolism as targets for new anthelmintics" for consideration at PLOS Neglected Tropical Diseases. As with all papers reviewed by the journal, your manuscript was reviewed by members of the editorial board and by several independent reviewers. In light of the reviews (below this email), we would like to invite the resubmission of a significantly-revised version that takes into account the reviewers' comments. 

The Reviewers have commented that the manuscript is overly long and speculative. We agree with them. In being highly speculative, and oversold, it loses some scientific credibility. This needs serious attention before the manuscript can be acceptable. In addition, the Reviewers have indicated that more statistical rigour is needed, either the need for more replication, controls or just statistical testing in places; as well as presentation of data mentioned as 'data not shown'. Such data can be included in Supplementary data. These points are also very important. There are many other aspects that the Reviewers have identified that need correction and which will improve the manuscript. There is much good original work and thinking in the manuscript, but it needs a major revision to meet publication standards for good science.

We cannot make any decision about publication until we have seen the revised manuscript and your response to the reviewers' comments. Your revised manuscript is also likely to be sent to reviewers for further evaluation.

Sincerely,

Roger K Prichard

Associate Editor

Aaron Jex

Deputy Editor

The Reviewers have commented that the manuscript is overly long and speculative. We agree with them. In being highly speculative, and oversold, it loses some scientific credibility. This needs serious attention before the manuscript can be acceptable. In addition, the Reviewers have indicated that more statistical rigour is needed, either the need for more replication, controls or just statistical testing in places; as well as presentation of data mentioned as 'data not shown'. Such data can be included in Supplementary data. These points are also very important. There are many other aspects that the Reviewers have identified that need correction and which will improve the manuscript. There is much good original work and thinking in the manuscript, but it needs a major revision to meet publication standards for good science.

Reviewer's Responses to Questions

**Key Review Criteria Required for Acceptance?**

**Methods**

-Are the objectives of the study clearly articulated with a clear testable hypothesis stated?

-Is the study design appropriate to address the stated objectives?

-Is the population clearly described and appropriate for the hypothesis being tested?

-Is the sample size sufficient to ensure adequate power to address the hypothesis being tested?

-Were correct statistical analysis used to support conclusions?

-Are there concerns about ethical or regulatory requirements being met?

Reviewer #1: No. As described below, the manuscript is difficult to read for flow and results. Some data require more replicates (only two or three data points are presented with no description of how independent they are). Statistical differences can not be significant with so few replicates in some comparisons.

Reviewer #2: The phylogenetic trees are inaccurate - there is no way that highly conserved proteins like AMPK that where present in the common ancestor of all the phyla you examine do not have a molecular phylogeny that mirrors the phylogenetic tree of the phyla those proteins were taken from (e.g. echinoderms and mammals are deuterostomes and should share a clade, likewise arthropods and nematodes). Fortunately, the trees don't really contribute anything to your conclusions and are not necessary for the important data, which are the sequence alignments, so you can just get rid of all those trees.

Reviewer #3: Fig 2. there should be biological replicates for rigour

Fig 3 and throughout- statistical analysis are needed where conclusions are drawn about presence/absence of effect

**Results**

-Does the analysis presented match the analysis plan?

-Are the results clearly and completely presented?

-Are the figures (Tables, Images) of sufficient quality for clarity?

Reviewer #1: No, again, we describe more below. The authors should consider using figures as presentation of one pathway or process. Some cross pathways and are more difficult to understand. Additionally, some alleles are explained and some are not (in figures). Parallel data presentations and conclusions would make the manuscript more interpretable.

Reviewer #2: 365 "We have thus identified two excellent targets for potential anthelmintics in the RQ synthesis

pathway: COQ-2 and TDO-2." I see the data for TDO-2 in figure 3 but where is the data for COQ-2?

Reviewer #3: line 480- 'data not shown'- the evidence should be provided

why in some cases is the KCN effect on motility shown and in others the recovery of motility following KCN exposure? Would it not be better to show both time-courses in all cases?

**Conclusions**

-Are the conclusions supported by the data presented?

-Are the limitations of analysis clearly described?

-Do the authors discuss how these data can be helpful to advance our understanding of the topic under study?

-Is public health relevance addressed?

Reviewer #1: In some cases, yes. As we describe below, the authors can mention undersampled and preliminary conclusions. Additionally and importantly, the C. elegans results are massively oversold to all STH and even all parasitic nematodes. The results do not require that degree of salemanship.

Reviewer #2: Yes

Reviewer #3: It is rather overly speculative in places, but nonetheless, I found it a very readable and intriguing account of the possibility that components of this pathway may present novel anthelmintic targets with good selective toxicity. 

the evidence for the involvement of the AMPK/mTOR pathway in sensing hypoxia is strong but could be substantiated by at least one rescue experiment

**Editorial and Data Presentation Modifications?**

Reviewer #1: C. elegans as a helminth. It really depends on who you ask. Some helminth parasitologists will bristle at this contention. Again, the results do not require that level of overselling.

Throughout some figures, the C. elegans convention is to italicize gene names with a hyphen between the three or four letter gene designation and the number. For example, dpy-10. The authors are not consistent about italics or the use of a hyphen in several figures.

Line 44: “resistance” should be clarified as “anthelmintic resistance”

Line 52: Suggest changing “some of this” to “some of these changes involve”

Line 86: Authors appear to be speaking more generally here; suggests making “host and STH” plural

Line 107-108: “previously described” Citation?

Table 1: There is no legend indicating what the colors/+ indicate

Line 281-283: Consider simplifying the sentence beginning with “Two of these…”

Line 290-291: Consider avoiding the absolute. “No high confidence target from our candidate genes”

Line 463: Suggest inserting a colon after including to make the list more distinct 

Line 539: Identify but what about synthesize? 

Figure 2a: Y-axis scale could be shortened to allow more room to visualize some of the smaller boxes. 

Figure 3d: Structure completely missing

Supplemental Figures: No fault of the authors, but printed versions of trees are difficult to read. The actual digital files are very clear. PLoS needs to update their figure rendering in the PDF builds! 

Supplemental Table 1: For PHX769 and 243, the color coding of mutations is not actually highlighted on the sequence change. Additionally, extend the column width for strain, as it appears strain names are pushed to two lines.

Reviewer #2: 'Helminth' is not a synonym for 'nematode'. A helminth is any wormlike parasite of interest to humans and may include, in addition to roundworms/nematodes, flatworms. C. elegans is not a parasite, and therefore not a helminth (line 83). Thus helminth is also not a phylogenetic classification. Comparing 'helminths' and annelids is not meaningful because annelids are a phylogenetic classification and 'helminths' are poly-phyletic. Since you seem to use helminth as a 

synonym for nematode, I would just use nematode, and explain to the reader that while the helminths you are interested in (the one for which C. elegans is a suitable model) are nematodes, not all nematodes are parasites, C. elegans is not a parasite, but it s a good model for parasitic nematodes because it shares with parasites the switch to rhodoquinone under anoxic conditions.

2. By 'hosts' you seem to mean mammals of the human or livestock variety so say that. Not all hosts are mammals - nematodes parasitize virtually every metazoan phyla - so it doesn't make sense to compare 'hosts' to molluscs (which are also hosts to many nematode parasites). (line 384 'These are the three sets of animals that are..', also 511)

Reviewer #3: The references need tidying up- for example spurious use of capitalisation- lack of italics where required

line 329 C. elegans, in italics

Line 32 'identify' not 'identity'

throughout C. elegans (in italics) not C.elegans

**Summary and General Comments**

Reviewer #1: The authors identify potential druggable targets in rhodoquinone-dependent metabolic pathway in C. elegans and maybe other nematode species. Rhodoquninone has previously been established as necessary for helminth (as well as molluscs and annelids) metabolism in anaerobic environments, such as those inhabited by many soil transmitted helminths within hosts. Because rhodoquinone metabolism is not used in humans or livestock species, anthelmintics targeting this metabolic pathway are possibly nematode-specific. The authors have identified nine specific targets necessary for Rhodoquinone metabolism, and because all nine have human orthologues, worked to determine which targets or residues within these enzymes are helminth-specific. Although the work presented in this manuscript is interesting and offers important findings for the study of potential anthelmintic targets, this paper suffers from overall issues with logical flow and overinterpretation of the data. This manuscript took us (a molecular geneticist and a parasitologist) several reads and many hours to break down the results for this review. It should not take that much work to understand a manuscript. Below are major points and suggestions for revision, followed by minor suggestions and specific edits.

(1) The authors overreach with the applicability of their results. Authors initially suggest focusing on targets that are differentially expressed between free-living and parasitic life stages. They demonstrate that these potential targets are effective at inhibiting RQ-dependent metabolism in C. elegans, a Clade V nematode. Two out of three example STHs presented in the introduction (Ascaris and whipworms) do not have free-living life stages in the environment. Differences between clades and life cycle biology limit the applicability of these targets to all STHs, and further testing would be necessary to confirm these findings for other clades. An example of such differences can be found in resistance to benzimidazoles in ascarids. Resistance to BZ in Clade V parasitic nematodes has been well documented to be associated with mutations at codons 167,198, and 200 in beta-tubulin. Kruken et al. found that in A. lumbricoides, efficacy of mebendazole was significantly reduced, but found no resistance-associated mutations across the beta-tubulin genes. Differences between clades may play important roles in how treatments that affect these potential targets will affect different species. 

There is also a lack of consideration of how pharmacokinetics may impact these proposed targets. The authors cite beta-tubulin as an anthelmintic target that is highly similar between parasites and hosts, but treatment remains parasite-specific and is safe for hosts. Although it is true that slight variation is thought to be the deciding factor in BZs not targeting host tubulins, pharmacokinetics within the host are also thought to play an important role. 

Although this work offers important new information on potential anthelmintic targets, data are overinterpreted to apply to all STHs, and assumes safety for hosts solely dependent on minor differences in proteins. The authors should do a substantial edit and reframing of their results. We believe that the C. elegans results (if improved as suggested below) can stand on their own and be important to present to the broader parasitology and NTD communities. The authors do not need to oversell their results to all STH and diverse nematode species.

(2) Significant edits in writing are suggested to improve clarity and readability. The Results section is filled with previously published data, better suited for an introduction. These paragraphs make the flow harder to interpret and new results are lost in the mix. Results should focus more on what was found in this work. Although these previous findings are necessary to understand what led to the current work, the Results section should be focused on the unpublished data. The background information can be included in the Introduction, first figure (as it is now), and in summary figure panels for each pathway/process.

The order of data and figure presentation also makes this paper difficult to follow. Results within one section and/or figure focus in and out of the different pathways. It is suggested that authors go layer by layer discussing the results more generally and then focusing on the specific targets in each pathway. For example, Figure 2 focuses on hif-1 results in part of A, and all of B, C, and D. The authors can remove the extra pathway data from Figure 1A and focus that figure and data discussion solely on hif-1. Other figures can be focused on specific pathways.

(3) Figures should be able to stand alone, and currently figures are difficult to interpret. It is suggested that authors first show the pathway with the steps and genes of interest being shown for each pathway. For example, first show a model with all components in panel A and then follow with the experimental results for subsequent panels For the audience of PLoS NTD, the authors need to provide a more understandable flow, data presentation in the figures, and writing. Otherwise, the manuscript will not have a significant impact on this community because of its lack of clarity.

(4) For results detailed in Figure 2a, there are not enough replicates to make strong conclusions. For most mutants tested, there are only 2-4 biological replicates. More data points are needed to make strong conclusions. Specifically, you can calculate a mean or median from two data points, but no statistician (or rigorous biologist) will ever believe the result. The authors should replicate the results or present the caveats of the under-replicated experiments. 

Minor Points:

(1) The authors should be more detailed in lines 41-47. The authors include levamisole with these other two drugs in the discussion but neglect to include it here. Resistance is widespread in livestock, but not in the nematode groups included here. To our knowledge, many of the studies in humans reporting lack of efficacy have potential confounding factors as briefly discussed in the Moser et al. cited here. It would be more accurate to say that in some regions, there are reports on lack of efficacy although resistance is not definitive. 

(2) Although not consistently used, there are a number of statements the authors use repeatedly but somewhat inaccurately. Authors refer to STH-specific active sites, but all work is done in the model nematode C. elegans, which is not an STH. Be more general with the language. Also what about non-STH parasitic helminths such as Brugia malayi? This would be worth at least briefly mentioning. As stated above, we believe the C. elegans results are interesting and important to present without overselling the relevance as broad STH treatments.

(3) In the discussion, the authors suggest that treatments targeting their identified enzymes should act to prevent RQ-dependent metabolism, leading to parasite death. Before this section, the authors were more dogmatic. For example, in line 66, “would thus kill”. The more speculative “should” works better in this context, as nothing can be known for sure until therapeutics that target these sites are actually used.

(4) All assays with C. elegans were done using the L1 life stage, after exposure to cyanide. How did this affect development in your strains? We would consider addressing the possibility of differences between the C. elegans life stages used here and those of the STHs infecting hosts in the Disussion. Parasitic nematodes develop within their eggs to later life stages. Have you considered possible differences in effects across these different life stages?

(5) As Figure 5 is showing negative results, it is suggested that it is moved to the supplemental figures, and indicated as negative results within the text.

(6) In Figure 6, Panel A is a gene model for coq-2 but not other potential targets have gene models. This specificity again distracts from the flow. We suggest adding the full gene model (not just four exons) to the supplement. As suggested above, Figure 6A could be a model of the pathway. In previous figures, genes were named simply with the mutant allele, and it is suggested this is made consistent with this figure.

Reviewer #2: The manuscript would greatly benefit from strict editing. The rationale for focussing on the rhodoquinone pathway is belabored at several points in the manuscript as is the role of the reversal of electrode transport (e.g. paragraph line 445). The discussion is nearly fully redundant with text in the results section. I would encourage the authors to instead use the discussion to address wider implications such as, why are Complex II drugs so easy to find (almost every drug screen in C. elegans finds one) even though the drugs are not screened under anoxic conditions? Don't your results mean that we should be doing drug screens in C. elegans under anoxic conditions to find anoxia-specific drugs? Just some examples off the top of my head.

Minor stuff

230 and 4.5 hours. Worms incubated in M9 buffer alone was... (were)

Fig 2. Are you sure high concentrations of KCN don't quench GFP - just a thought.

Reviewer #3: This is a very interesting paper of considerable breadth investigating hypoxic metabolism in C. elegans. It is rather overly speculative in places, but nonetheless, I found it a very readable and intriguing account of the possibility that components of this pathway may present novel anthelmintic targets with good selective toxicity. 

I disagree with the statement on line 631-line 636 – cite ivermectin as example of anthelmintic that targets invertebrate unique ion channel GluCl. also monepantel targets a unique nicotinic channel.

It is rather long and somewhat repetitious in places and could be edited for brevity.

PLOS authors have the option to publish the peer review history of their article (what does this mean?). If published, this will include your full peer review and any attached files.

Reviewer #1: No

Reviewer #2: No

Reviewer #3: No
---

## [Decision Letter · Decision Letter 1]

11 Nov 2021

Dear Prof. Fraser,

We are pleased to inform you that your manuscript 'Identification of enzymes that have helminth-specific active sites and are required for Rhodoquinone-dependent metabolism as targets for new anthelmintics' has been provisionally accepted for publication in PLOS Neglected Tropical Diseases.

Best regards,

Aaron R. Jex

Deputy Editor

Aaron Jex

Deputy Editor

Reviewer's Responses to Questions

**Key Review Criteria Required for Acceptance?**

**Methods**

-Are the objectives of the study clearly articulated with a clear testable hypothesis stated?

-Is the study design appropriate to address the stated objectives?

-Is the population clearly described and appropriate for the hypothesis being tested?

-Is the sample size sufficient to ensure adequate power to address the hypothesis being tested?

-Were correct statistical analysis used to support conclusions?

-Are there concerns about ethical or regulatory requirements being met?

Reviewer #2: (No Response)

Reviewer #3: (No Response)

**Results**

-Does the analysis presented match the analysis plan?

-Are the results clearly and completely presented?

-Are the figures (Tables, Images) of sufficient quality for clarity?

Reviewer #2: (No Response)

Reviewer #3: (No Response)

**Conclusions**

-Are the conclusions supported by the data presented?

-Are the limitations of analysis clearly described?

-Do the authors discuss how these data can be helpful to advance our understanding of the topic under study?

-Is public health relevance addressed?

Reviewer #2: (No Response)

Reviewer #3: (No Response)

**Editorial and Data Presentation Modifications?**

Reviewer #2: (No Response)

Reviewer #3: (No Response)

**Summary and General Comments**

Reviewer #2: (No Response)

Reviewer #3: The authors have revised the manuscript broadly according to the recommendations.

PLOS authors have the option to publish the peer review history of their article (what does this mean?). If published, this will include your full peer review and any attached files.

Reviewer #2: No

Reviewer #3: No

---

## [Editor Report · Acceptance letter]

22 Nov 2021

Dear Prof. Fraser,

We are delighted to inform you that your manuscript, "Identification of enzymes that have helminth-specific active sites and are required for Rhodoquinone-dependent metabolism as targets for new anthelmintics," has been formally accepted for publication in PLOS Neglected Tropical Diseases.

Best regards,

Shaden Kamhawi

co-Editor-in-Chief

Paul Brindley

co-Editor-in-Chief
